

# Evaluation of the IAGOS-Core GHG Package H2O measurements during the DENCHAR airborne inter-comparison campaign in 2011

Annette Filges[1], Christoph Gerbig[1], Chris W. Rella[2], John Hoffnagle[2], Herman Smit[3], Martina Krämer[4], Nicole Spelten[4], Christian Rolf[4], Zoltán Bozóki[5], Bernhard Buchholz[6], Volker Ebert[6]

[1]Max Planck Institute for Biogeochemistry (MPI-BGC), Jena, 07747, Germany
[2]Picarro, Inc., Santa Clara, CA 95054, USA
[3]Research Centre Jülich, Institute for Energy and Climate Research Troposphere (IEK-8), Jülich, 52428, Germany
[4]Research Centre Jülich, Institute for Energy and Climate Research Stratosphere (IEK-7), Jülich, 52428, Germany
[5]Hungarian Academy of Sciences (MTA) and University of Szeged (SZTE) Research Group on Photoacoustic Spectroscopy,
Szeged, H-6720, Hungary
[6] Physikalisch-Technische Bundesanstalt (PTB), Braunschweig, 38116, Germany

*Correspondence to*: Annette Filges (annette.filges@bgc-jena.mpg.de)

**Abstract.** As part of the DENCHAR (Development and Evaluation of Novel Compact Hygrometer for Airborne Research) inter-comparison campaign in North-Germany in 2011, a commercial cavity ring-down spectroscopy (CRDS) based gas analyzer (G2401-m, Picarro Inc.,US) was installed on a Learjet to measure atmospheric water vapor, $CO_2$, $CH_4$ and CO. The CRDS components were identical to those chosen for integration aboard commercial airliners within the IAGOS (In-service Aircraft for a Global Observing System) project. Thus, the campaign allowed for an initial assessment validation of the long-

term IAGOS water vapor measurements by CRDS against reference instruments with a long performance record (Fast In-situ Stratospheric Hygrometer (FISH), CR-2 frost point hygrometer (Buck Research Instruments L.L.C., US), both operated by research centre Juelich).

The inlet system, a 50 cm long 1/8" FEP-tube connected to a Rosemount TAT housing (model 102BX, deiced) installed on a window plate of the aircraft, was designed to eliminate sampling of larger aerosols, ice particles, and water droplets, and

provided additional ram-pressure. In combination with a low sample flow of 100 sccm, corresponding to a 4 second response time, this ensured a fully controlled pressure in the sample cell of 186.65 hPa (140 Torr) throughout the aircraft altitude operating range up to 12.5 km without the need of an upstream sampling pump. This setup ensures full compatibility with the deployment of the analyzer within IAGOS.

For the initial water calibration of the instrument it was compared against a dew point mirror (Dewmet TDH, Michell

instruments Ltd., UK) in the range from 0.7 to 2.5 % water vapor mole fraction. During the inter-comparison campaign the analyzer was compared on ground against a dew point hygrometer, which is used for calibrating the reference instrument FISH, in the range from 2 to 600 ppm. Furthermore, a new independent calibration method, based on the dilution effect of water vapor on $CO_2$, was tested.

Comparison of the in-flight data against the reference instruments showed that the analyzer is reliable and has a good long-

term stability. The flight data suggest a conservative precision estimate for measurements made at 0.4 Hz (2.5 seconds





measurement interval) of 4 ppm or 5 % (relative) (whichever is greater) for $H_2O$ < 100 ppm, and 5 % (relative) or 30 ppm (whichever is smaller) for $H_2O$ > 100 ppm. Accuracy of the CRDS instrument was estimated, based on laboratory calibrations, as 1 % (relative) for the water vapor range from 2.5 % down to 0.7 %, than increasing to 5 % (relative) at 50 ppm water vapor. Accuracy at water vapor mole fractions below 50 ppm was difficult to assess, as the reference systems

5  suffered from lack of data availability.



# 1 Introduction

Water vapor is a crucial factor for various atmospheric processes, weather, and climate. It is the most important greenhouse gas (Kiehl and Trenberth, 1997) and shows strong feedback to changes in the climate system (Dessler, 2008). Especially in the upper troposphere and lower stratosphere (UTLS) the amount of water vapor has large impact on the radiative balance of

the atmosphere (e.g. Smith et al., 2001; Forster and Shine, 2002; Solomon et al., 2010). Since there are only sparse measurement data of sufficient quality in the UTLS and, as emphasized in Solomon et al. (2010), prognostic models have difficulties to represent this region well, uncertainties in chemistry, transport processes, and trace gas composition are relatively large, which influences significantly the estimation of e.g. radiative effects (Riese et al., 2012; Kunz et al., 2013).

Water vapor observations covering the whole troposphere and at least lower parts of the stratosphere are achieved mainly by

instruments based on balloons, aircraft or satellites, and from ground based remote sensing techniques. The longest measurement time series was started in 1980 in Boulder (Colorado, US) with balloon-borne frost point hygrometers (Oltmans et al., 2000; Hurst et al., 2011). First long-term global satellite data were obtained in the mid-1980s as part of the Stratospheric Aerosol and Gas Experiment II (SAGE II) (Rind et al., 1993), followed by observations from e.g. the Halogen Occultation Experiment (HALOE) (Harries et al., 1996), the Aura Microwave Limb Sounder (MLS) (Read et al., 2007;

Lambert et al., 2007), as well as the Michelson Interferometer for Passive Atmospheric Sounding (MIPAS) (Milz et al., 2005; von Clarmann et al., 2009) and the Scanning Imaging Absorption Spectrometer for Atmospheric Chartography (SCIAMACHY) (Rozanov et al., 2011; Weigel et al., 2015), both aboard ENVISAT (Environmental Satellite). The main drawbacks of satellite data and remote sensing observations from ground (e.g. Schneider et al., 2006) are their insufficient spatial resolution in the troposphere and lower stratosphere and disturbances of the measurements by clouds. On the other

hand reliable radiosonde water vapor data up to stratospheric heights, e.g. from the GCOS (Global Climate Observing System) Reference Upper-Air Network (GRUAN) (Dirksen et al., 2014), as well as data sets from research aircraft are quite limited in time and space.

The use of commercial aircraft as cost-efficient platforms for dedicated instruments can at least partially bridge this gap providing regular measurements in the UTLS along major flight routes. For example, five Airbus A340 passenger aircraft

were equipped with capacitive humidity sensors from 1994-2014 as part of the MOZAIC (Measurement of Ozone and Water Vapor by Airbus In-Service Aircraft) project (Marenco et al., 1998). The acquired data set is crucial for the study of chemical and dynamic processes in the upper troposphere and lower stratosphere (e.g. Gierens et al., 1999).

However, accurate and reliable airborne measurements of atmospheric water vapor are still a challenge. The large range from mole fractions of several percent at the ground to only a few parts per million (ppm = $\mu$mol*mol$^{-1}$) in the stratosphere and

the highly variable structures of water vapor in the atmosphere are demanding for analyzers regarding accuracy and time response.

Kley et al. (2000) gives a detailed summary of the most important water vapor instruments used onboard aircraft. Beneath frost point hygrometers (e.g. Vömel et al., 2007; Vömel et al., 2016; Hurst et al., 2011; Hall et al., 2016) these are mainly





Lyman-α hygrometers, based on fluorescence techniques, for example the Harvard Water Vapor instrument (HWV) (Weinstock et al., 2009) and the Fast In Situ Stratospheric Hygrometer (FISH) (Zöger et al., 1999; Meyer et al., 2015). More recently infrared absorption spectrometers like the Jet Propulsion Laboratory Laser Hygrometer (JLH) (May, 1998), the Integrated Cavity output Spectrometer (ICOS) (Sayres et al., 2009), or the Hygrometer for Atmospheric Investigations (HAI)

(Buchholz et al., 2017), and the Atmospheric Ionization Mass Spectrometer (AIMS) (Kaufmann et al., 2016) have been deployed. The central problematic of all these different types of analyzers are the remaining, unexplained discrepancies between water vapor measurements in the range below 10 ppm (e.g. Kley et al., 2000; Vömel et al., 2007; Weinstock et al., 2009). While the instruments compare well during static experiments (Fahey et al., 2014), they disagree significantly during airborne inter comparisons in the UTLS. For the recent Mid-latitude Airborne Cirrus Properties Experiment (MACPEX) in

2011 Rollins et al. (2014) estimated the differences to be of the order of 20 % at water vapor mixing ratios of 3-4 ppm, whereas the measurement uncertainties of the instruments account only for 5-10 %.

In this context the DENCHAR (Development and Evaluation of Novel Compact Hygrometer for Airborne Research) project was initiated by the European Facility for Airborne Research (EUFAR) to support the development and characterization of novel or improved compact airborne hygrometers for different airborne applications within EUFAR, including investigation

of the sampling characteristics of different gas/ice inlets (cf. Tátrai et al., 2015). As part of the DENCHAR inter-comparison flight campaign in Hohn (Germany) in May-June 2011, a commercial cavity ring-down spectroscopy (CRDS) gas analyzer (G2401-m, Picarro Inc.,US), measuring $CO_2$, $CH_4$, CO and water vapor, was tested and compared against well-established reference hygrometers and newly developed water vapor instruments. Four flights with a Learjet 35A took place in an area between North-Germany and South-Norway, and North-Poland and the North Sea respectively, reaching altitudes up to 12.5

km, hence covering also the lower stratosphere. As reference instruments served the Learjet version of the Fast In-situ Stratospheric Hygrometer (FISH) and a CR-2 frost point hygrometer (Buck Research Instruments L.L.C., Boulder, US), both operated by the research centre Jülich.

The same CRDS analyzer and corresponding inlet system components are scheduled for deployment onboard passenger aircraft within the IAGOS (In-service Aircraft for a Global observing System) project (Filges et al., 2015). IAGOS was

launched in 2005 as the successor program of MOZAIC, but with modernized instrumentation and enhanced measurement capabilities (Volz-Thomas et al., 2009; Petzold et al., 2015). The current fleet of IAGOS-equipped aircraft as well as the spatial coverage of all flights can be found at the IAGOS-database (www.iagos.org). It is planned to equip five IAGOS aircraft with the CRDS system, as "IAGOS-core Greenhouse Gas (GHG) package", in the next four years. Since the MOZAIC humidity device (Helten et al., 1998, Smit et al., 2008, Smit et al., 2013) is deployed on each IAGOS aircraft, this

combination provides the unique opportunity to compare both instrument types under the same conditions over a long-time period. IAGOS water vapor measurements include regular in situ data in the sensible UTLS region and vertical profiles of $H_2O$ in the troposphere and lower stratosphere for major parts of the globe. They are essential for validation of remote sensing based observations from satellites and ground, are needed for improving the performance of climate models and weather forecasts, and can be used for climate trend studies.



This paper presents the water vapor measurements made with the CRDS system during the DENCHAR inter-comparison flight campaign in 2011. The flight data are validated against reference instruments with a long performance record (FISH (Meyer et al., 2015) and CR-2 Cryogenic Aircraft Hygrometer (Buck Research Instruments L.L.C., US, www.hygrometers.com)) to evaluate the water vapor measurements made by the CRDS instrument. Furthermore, the

analyzer was calibrated with the help of different hygrometers and a new independent calibration method was tested. The corresponding results are analyzed and discussed regarding the feasibility of the different methods for the long-term operation of the analyzer within the IAGOS project.

The measurement system is introduced in Sect. 2, followed by an overview of the water vapor calibration approaches in Sect.3. Results from the flight tests, including comparison with the reference instruments, are presented in Sect. 4. Section 5

concludes the paper.

## 2 The measurement system

The measurements were conducted by a G2401-m wavelength scanned cavity ring-down spectroscopy analyzer from Picarro Inc. (US) (CFKB2004), which simultaneously measures $CO_2$, $CH_4$, CO and water vapor (Crosson, 2008; Chen et al., 2010).

The CRDS technique determines the mole fraction of a gas using the decay time of light intensity ("ring-down time") due to

absorption by the gas. Laser light of a specific set of wavelengths is injected into a mirrored sample cell (the "cavity", 35 cm³, effective optical path length 15-20 km), which is flushed with the sample gas. When the light intensity reaches a predetermined threshold, the laser is turned off, after which the optical energy in the cavity decays with a characteristic exponential time constant of the light intensity in the cavity (the ringdown). The total absorption of the cavity (including both the absorption of the gas and the loss of the mirrors) is calculated directly from the exponential time constant. By tuning

the wavelength of the laser, a specific spectral line of a species is scanned and analysis of the obtained spectrogram provides the peak height, which at constant pressure and temperature is proportional to the mole fraction of the species.

The analyzer uses selected spectral lines in the infrared for the measurements: at 1603 nm for $^{12}C^{16}O_2$, at 1651 nm for $^{12}CH_4$ and $H_2^{16}O$, and at 1567 nm for $^{12}C^{16}O$.

To minimize impact on gas density and spectroscopy, pressure and temperature in the sample cell are kept constant. Pressure

in the sample cell is controlled to 186.65 ± 0.04 hPa (140 Torr) using a proportional valve ("inlet valve") upstream of the cell, and the temperature is kept at 45 ± 0.02 °C. Gas flow through the sample cell was controlled at 100 sccm with the help of a fixed needle valve, acting as flow-restricting orifice, downstream of the sample cell and upstream of the pump. Thus, the flow rate was independent on ambient, respectively cabin pressure.

To protect the sample cell from contamination, two filters (Wafergard II F Micro In-Line Gas Filters, Entegris Inc.) are

located in the sample line upstream of the sample cell. They also ensure thermal equilibration of the sample gas, as they are kept at the same temperature as the sample cell.



Each species was measured once every 2.5 seconds. The physical exchange time of the sample cell is 3.6 seconds (volume = 35 cm³, sample flow = 100 sccm, pressure = 186.65 hPa, sample temperature = 45 °C), ensuring that the ambient air was continuously sampled given the shorter measurement interval of 2.5 seconds.

The instrument was equipped with a 50 cm long inlet line (3.18 mm (1/8") OD, 1.58 mm (1/16") ID, Fluorinated Ethylene Propylene (FEP) tube), which was connected to a Rosemount Total Air Temperature (TAT) housing (model 102B; Stickney et al., 1994) mounted on a window plate of the Learjet. The Rosemount probe acts as a virtual impactor since the inlet line is pointed orthogonal to the airflow through the housing (see Fig. 1), and thus prevents from sampling larger aerosols (larger than about 2 µm), ice particles, and water droplets (Volz-Thomas et al., 2005; Fahey et al., 2001, Smit et al., 2013). Furthermore, it provides positive ram-pressure due to the reduction of the air velocity. This additional positive ram-pressure, together with the low sample gas flow of 100 sccm and the relatively short inlet line, ensured operation of the instrument throughout the aircraft altitude operating range up to 12.5 km without an upstream sampling pump. Diffusion effects of water vapor in the inlet line are minimal, given the short residence time of the sample gas, the small inner surface area, the small differences in humidity between cabin and ambient air, and the low permeability of FEP. The sample flow was exhausted into the cabin of the aircraft.

The CRDS analyzer and the inlet system components are identical to those chosen for integration aboard commercial airliners within the IAGOS (In-service Aircraft for a Global observing System) project. This setup ensures full comparability with the deployment of the IAGOS-core Greenhouse Gas package (Filges et al., 2015).

**3 Calibration**

In contrast to calibration of the $CO_2$, $CH_4$ and CO measurements, for which traceability to the World Meteorological Organization (WMO) primary scales is ensured by measurement of gas standards traceable to the primary scale, calibration of the water vapor measurements of the instrument is not as straight forward. There is no globally valid primary scale, but several national standards exist (WMO, 2012, see Part I, Chapter 4). Calibration of an instrument is done by means of other hygrometers that are traceable to one of the national standards, often gravimetric hygrometers. In the following different calibration methods for the CRDS analyzer are presented and compared.

**3.1 Methods**

**3.1.1 Offset Correction**

Prior to calibration of the CRDS analyzer against e.g. a dew point mirror an offset correction is required to improve measurements at low water levels (<0.1 %). This offset correction can be estimated by measuring dried ambient air from a high-pressure tank. At the MPI-BGC Gaslab the tanks (volume: 50 liters) were filled with air, which was dried using magnesium perchlorate ($Mg[ClO_4]_2$). The dew point of the air is around -75 °C corresponding to 2.4 ppm water vapor.



### 3.1.2 Dew Point Mirror

The factory calibration of the Picarro Inc. CRDS analyzer consists of two parts: A self-broadening correction and a comparison with a dew point mirror.

Water vapor mole fraction is calculated using the peak height of the selected water absorption line. In this process self-
broadening effects must be taken into account, which broaden the line shape and hence decrease the peak height, as the water vapor level increases. To avoid an underestimation of the water vapor mole fraction, a quadratic correction is implemented in the Picarro analyzer (Rella, 2010):

$$H_2O_{corrected} = H_2O_{reported} + 0.02525 * H_2O_{reported}^2 \qquad (1)$$

Here, $H_2O$ is water vapor mole fraction in %. In 2009 a G1301-m CRDS instrument from Picarro, measuring $CO_2$, $CH_4$ and
$H_2O$, was calibrated at MPI-BGC Jena against a dew point mirror (Dewmet TDH, Cooled Mirror Dewpointmeter, Michell Instruments Ltd., UK, referenced to National Institute of Standards and Technology (NIST) primary scale) in the range from 0.7 to 3.0 % water vapor mole fraction (Winderlich et al., 2010). The calibration constant obtained in this experiment was transferred to all greenhouse gas CRDS instruments manufactured by Picarro Inc.  (Rella, 2010):

$$H_2O_{calibrated} = 0.772 * H_2O_{corrected} = 0.772 * \left( H_2O_{reported} + 0.02525 * H_2O_{reported}^2 \right) \qquad (2)$$

This calibration transfer from one to all other instruments is possible since their water vapor measurements agree within a sufficient range and are stable over time as shown by Rella et al. (2013). Here, three different analyzers (models G2401-m and Envirosense 3000) were compared at different times against one selected standard instrument (CFADS37, model G1301-m). One of the comparisons was repeated after more than three years. All results ($H_2O_{analyzer} - H_2O_{CFADS-37}$) lie within a range of ±125 ppm for water vapor mole fractions ranging from around 3 ppm to 3 %. Hence, a good transferability and long-term
stability of the analyzers water vapor measurements can be assumed.

In order to examine the robustness of the 2009 calibration it was repeated in 2013 using a G2401-m analyzer (CFKBDS2003) in comparison to the identical dew point mirror (Dewmet TDH, Michell Instruments Ltd.) that was used for the calibration in 2009. Both instruments measured simultaneously dried ambient air from a high-pressure tank, which was humidified by a dew point generator (Li-610 from Li-Cor) to specific water levels between 2°C and 20°C dew point.
During the 2009 calibration the dew point mirror measurement was based on its original calibration, conducted by the manufacturer against test equipment traceable to the NIST primary standard in the end of 2000. In 2010 the dew point mirror was recalibrated by the manufacturer, however, no information was given on how the calibration factors changed. Another calibration by the manufacturer in 2014, shortly after the 2013 comparison of the CRDS instrument and the dew point mirror, showed no drift beyond the uncertainty range of the dew point mirror (given by manufacturer: 0.2°C at +20°C dew
point, increasing linearly to 0.4°C at -60°C dew point) compared to the calibration in 2010.



### 3.1.3 Calibration Bench for FISH-Instrument

During the DENCHAR inter-comparison campaign in 2011 the CRDS analyzer CFKB2004 was compared against the laboratory calibration bench used regularly for calibration of the reference instrument FISH. This calibration bench consists of a humidifier, a mixing unit to mix dry and humid air, and a reference water vapor instrument, the MBW Dew Point

instrument (model K-1806/DP30-SHSX- III, MBW Elektronik AG, Switzerland, www.mbw.ch) (Meyer et al., 2015). For the comparison the CRDS instrument was connected to the calibration bench via a 3 meter long 1/8" FEP-line. Since the calibration bench provided a flow of about 3500 sccm an open-split was installed in front of the FEP-line to allow the CRDS analyzer to sample at its low flow rate of 100 sccm. During the comparison four humidity steps covering the range of 2 to around 600 ppm were measured. Maximum uncertainty of the calibration bench is given as ±4 % (relative) by Meyer et al.

10   (2015).

Due to the low sample flow (100 sccm) through the analyzer and the large difference in water vapor mole fraction between the measured air and the outside air during the comparison, permeation of water vapor through the FEP-tube (3 m length in the calibration setup) has to be considered. To provide information from which a correction factor for the permeation effect could be determined, a dry tank air stream (~2 ppm water vapor mole fraction) at different flow rates (100 sccm and 3500

sccm) was provided through the FEP-tube. Assuming that for a flow of 3500 sccm the permeation is negligible the correction factor was computed as the difference in the calibrated CRDS $H_2O$ mole fraction between these two measurements.

### 3.1.4 Calibration by $CO_2$ dilution effect

In addition to the standard calibrations by different hygrometers a new, and completely independent calibration method was tested, which takes advantage of the high precision $CO_2$ measurements by the CRDS analyzer. Specifically, the dilution

effect of water vapor on the $CO_2$ mole fraction is used: If water vapor is added to dry air, while total pressure and temperature of the gas remain unchanged, the mole fractions of the residual air components decrease. The mole fraction of $CO_2$ in dry air is

$$X_{CO_2}^{dry} = \frac{n_{CO_2}}{n_{air}+n_{CO_2}} \ ,$$ (3)

whereas the $CO_2$ mole fraction in wet air is given as

$$X_{CO_2}^{wet} = \frac{n_{CO_2}}{n_{air}+n_{CO_2}+n_{H_2O}}$$ (4)

Together with the wet air mole fraction of water,

$$X_{H_2O}^{wet} = \frac{n_{H_2O}}{n_{air}+n_{CO_2}+n_{H_2O}} \ ,$$ (5)

Eqs. (3) and (4) yield:

$$\frac{X_{CO_2}^{wet}}{X_{CO_2}^{dry}} = 1 - X_{H_2O}^{wet}$$ (6)





Thus, the amount of water vapor in air is directly linked to the ratio of the $CO_2$ wet and dry air mole fraction of the air. However, the measured $CO_2$ mole fraction of the CRDS instruments in wet air is not only influenced by the dilution effect, but also through pressure broadening effects of the water vapor. To separate both effects the measurement software of the analyzer was modified to allow for a fine scan of the $CO_2$ and water vapor absorption line. While the peak height, which is

normally used for the measurement, is influenced by both effects, the peak area gets only changed by the dilution effect. The fine scan, combined with spectral models and fitting procedures optimized for this purpose, provides the peak areas with sufficient precision.

To test the concept pressurized zero air with 3000 ppm $CO_2$ from a high pressure tank was split into two paths, as can be seen in Fig. 2. The air in one path was humidified in a bubbler. Afterwards the dry and wet gas stream were recombined and

then measured by a CRDS-analyzer (CFADS2196, model G2301) in fine scan mode. With the help of mass flow controllers in both paths the water concentration of the combined stream could be varied without changing the $CO_2$ dry mole fraction by changing the flow of each path while the total flow was kept constant. The measurements alternated between the water line and the $CO_2$ line, whereby it required about 1.3 s to make one pair of measurements. Since the pressure and temperature in the sample cell were kept constant the measured peak areas were proportional to the mole fractions:

$$X_i = C_i * A_i \tag{7}$$

Where A is the peak area and C the proportionality or "calibration" factor. With Eq. (6) this yields:

$$A_{CO_2}^{wet} = A_{CO_2}^{dry}\left(1 - C_{H_2O} * A_{H_2O}\right) \tag{8}$$

If the measured area of the $CO_2$ line is plotted as a function of the measured area of the water line, the calibration factor for water vapor $C_{H2O}$ is just the ratio of the slope and the intercept. The scan of the water line provided also the conventional

water vapor measurement using the peak height of the absorption line, which allowed for comparing the two water vapor estimates.

### 3.2 Results and Discussion

### 3.2.1 Dew Point Mirror

Plot (a) in Fig. 3 shows the self-broadening and offset corrected, but uncalibrated, water vapor mole fraction measured by the

Picarro CRDS instrument (in the following referred to as "$H_2O_{uncalibrated}$") against the measurements of the dew point mirror (using its factory calibration from 2010, which was confirmed in 2014) during the comparison in 2013. The Dewmet measurements were converted from dew point to wet air mole fraction based on the Goff-Gratch equation (Goff, 1957). The corresponding fit can be seen as blue line, the grey line indicates the calibration curve of the 2009 experiment (Eq. (2)). The uncertainty of the fitted slope is composed of the fitting error and the uncertainty of the dew point mirror. Uncertainty of the

Dewmet is given by the manufacturer as 0.2°C at +20°C dew point, increasing linearly to 0.4°C at -60°C dew point. In order to check the linearity of the CRDS instrument the CRDS and Dewmet data were also fitted using a quadratic fit. The slope of the quadratic fit was determined as 0.807 ± 0.011, which agrees well with the slope of the linear fit (0.802) taking account of





the uncertainty range. The fit coefficient of the quadratic term, determined as 0.0024 ± 0.0021, is not significant. Thus, the CRDS analyzer can be considered as linear.

Figure 3 b) shows the difference between the 2009 and the 2013 calibration. The error bars demonstrate the uncertainty range, which comprises the dew point mirror uncertainty during the 2009 as well as during the 2013 experiment. The relative

difference of the two calibrations increases from 2.2 % (relative) at 0.8 % water vapor mole fraction up to 4.4 % (relative) at 2.5 %, indicating significant differences for water vapor mole fractions above about 1 %.

This difference between the 2009 and the 2013 calibrations is much larger than the uncertainties of the instruments and the calibration transfer give reason to expect. The largest source of uncertainty is the dew point mirror with 1.3 % (rel.)

uncertainty. Repeatability of the CRDS analyzer is given as <14 ppm by the manufacturer. Uncertainty of the calibration transfer between different Picarro analyzers, which has to be considered since the two calibrations were done with different CRDS instruments, is <125 ppm (or 0.5% relative at 2.5% water vapor mole fraction) (Rella et al., 2013). Since the difference between the two calibrations is up to 4.4 % (rel.) the dew point mirror, the Picarro analyzer or both instruments must have been drifting.

In order to test the stability of the water vapor measurements of the CRDS analyzer, the CFKBDS2003 instrument was compared to another Picarro CRDS analyzer (CFADS37, model G3101-m) once in 2011 and again in 2014. During the experiments both instruments measured in parallel pressurized, dried ambient air from a high-pressure tank, which was humidified by a dew point generator (Li-610 from Li-Cor) to specific water levels between 2°C and 20°C dew point. Plot (a) in Fig. 4 shows the result of the 2014-comparison (black points). The blue line is the quadratic fit of the 2011-comparison.

The difference (plot (b)) between the two experiments are <0.3 % (rel.). Since it is unlikely that both instruments drifted in the same way, this strongly suggests that the CFKBDS2003 analyzer did not drift significantly in the three years between the two comparisons against the dew point mirror.

This conclusion together with the calibration history of the dew point mirror (see Sect. 3.1) suggests that the large

differences between the two calibrations of the CRDS instrument in 2009 and 2013 are caused by drift of the dew point mirror calibration. During the 2009 experiment the dew point mirror was not calibrated well enough and thus, only the results of the 2013 experiment, corresponding to a calibration factor of 0.802 for the CRDS water vapor measurements, are reliable. Accuracy of the calibration is limited by the uncertainty range of the dew point mirror (1.3 % (rel.)). For water vapor levels <0.7 % the calibration is only extrapolated based on the measurements between 0.7 − 2.5 %, which has to be

accounted for in the uncertainty estimate.

### 3.2.2 Calibration Bench for FISH-Instrument

Plot (a) in Fig. 5 shows the result of the comparison between the CRDS analyzer and the FISH calibration bench, during which four different water levels in the range 2-600 ppm were measured. The water vapor measurements of the CRDS





analyzer are offset corrected and calibrated according to the 2013 dew point mirror comparison (calibration factor = 0.802 ± 0.010), and additionally corrected for another 3.5 ppm resulting from permeation of water vapor from air surrounding the 3 m FEP inlet line. Subsequently, the CRDS measurements were converted from wet to dry air mole fractions according to:

$$H_2O_{dry} = \frac{H_2O_{wet}}{1 - H_2O_{wet}} \qquad (9)$$

A linear fit of the data (blue line) shows that the dew point mirror-calibration of the CRDS was within 3 % of the FISH calibration bench and showed an offset of 12.2 ppm. Uncertainties of the fit coefficients (slope: ±0.04, offset: ±0.5 ppm) were estimated assuming a worst case scenario including 4 % (rel.) bias of the FISH calibration bench and 1.3 % (rel.) uncertainty of the dew point mirror calibration. The residuals (difference between the Fish calibration bench and the fit), which can be seen in plot (b) are small compared to the uncertainty range of the FISH calibration bench of 4 % (rel.)

indicated by the error bars.

The CRDS analyzer and the FISH calibration bench agree within 3 % (rel.) in the water vapor range up to 600 ppm. This confirms that extrapolation of the dew point mirror calibration to water vapor levels below 0.7 % is appropriate, at least within the uncertainty of 4 % (rel.) assumed for the calibration bench. Regarding the offset of 12.2 ppm it has to be considered that the measured air at the lowest water level, which has the largest effect on the estimation of the offset, was

perhaps not completely in equilibrium with the inner surface of the connection line between the CRDS instrument and the calibration bench and the tubing inside the analyzer. If the water vapor mole fraction in the gas stream decreases water molecules adsorbed at the surface are released until a new equilibrium with the air is reached. Due to the large length of the connection line (3 m) the inner surface is relatively large and thus, the balancing process takes relatively long time. For the higher measured water levels the balance is reached faster and furthermore, except for the highest measured level, memory

effects were canceled out by measuring the water level twice: once from low to high, and once going from high to low mole fractions. The differences in the two measurement sequences are smaller than 1 ppm.

### 3.2.3 Calibration by CO$_2$ dilution effect

Figure 6 a) shows the comparison of the water vapor mole fraction determined with the help of the CO$_2$ dilution method and the conventional water vapor measurements, which are offset corrected and calibrated according to the 2013 dew point

mirror comparison (calibration factor = 0.802 ± 0.010), during the fine scan experiment. A linear fit of the data (blue line) indicates a bias of 2.9 % (rel.) of the dilution based water vapor compared to the dew point mirror calibration and an offset of 16.1 ppm. The uncertainty estimates (slope: ±0.013, offset: ±1.6 ppm) are based on the uncertainty of the dew point mirror calibration (1.3 % rel.). Residuals can be seen in plot (b) of Fig. 6.

The water vapor mole fraction calculated with the CO$_2$ dilution method and the conventional water vapor measurements calibrated according to the 2013 dew point mirror comparison agree within 2.9 % (rel.) in the water vapor range from 300

ppm to 2.7 %. The residuals (difference between water vapor mole fraction based on dilution method and the linear fit) are small, but show a slight systematic shape depending on the water vapor level. An offset was determined as 16.1 ppm,



however the lowest measurement was made at around 300 ppm and thus, the offset is based on extrapolation. Higher scatter in the residuals at low water vapor (<0.25 %) might indicate a different behavior for this range. Hence, the estimated error of 1.6 ppm for the offset is likely unrealistic.

Estimating the uncertainty of the $CO_2$ dilution method is not straightforward. The repeatability of the peak area
measurements accounts only for less than 0.1 % (rel.) uncertainty, whereas systematic errors can have a larger influence on the accuracy. One potential error is the direct spectroscopic interference of either water on $CO_2$ or vice versa, which was tried to exclude by careful selection of the used absorption lines and detailed spectral models. To check for remaining influences an additional test was conducted: Since a direct spectroscopic interference would affect the measurements differently for different $CO_2$ concentrations, the fine scan experiment was repeated with 400 ppm $CO_2$ instead of the original
3000 ppm $CO_2$. Unfortunately, the pressurized zero air with 400 ppm $CO_2$ also contained 2 ppm of methane, whereas the 3000 ppm $CO_2$ air was pure $CO_2$ in zero air. Thus, a neighbouring methane absorption line had to be considered, which added another variable to the analysis. In future experiments this should be excluded by preparing a set of high pressure tanks of exactly the same air composition but different $CO_2$ concentrations. The calibration constant $C_{H2O}$ of the water measurements (see Eq. (7) and (8)) measured for 400 ppm $CO_2$ was 0.6 % (rel.) larger than the calibration constant measured
for 3000 ppm $CO_2$. Another systematic error can arise if the spectroscopic models and fitting procedures do not perfectly account for the changes in the absorption line shapes during varying water vapor mole fractions. For this experiment the absorption line shape model was carefully tested over the range of conditions in the analyzer, and it was found that the corresponding error can be neglected compared to the other sources of uncertainty.

Recently, a potentially serious source of systematic error regarding the pressure control in the sample cell was discovered:
Observations suggest that the pressure sensor is sensitive to water vapor and thus, the pressure in the sample cell is stabilized to a humidity dependent value instead of the fixed 186.65 hPa (= 140 Torr). To assess the quantitative effect of such an incorrect pressure reading we assume the pressure in the measurement cell to be

$$p = p_0 + \Delta p \,, \tag{10}$$

where $p_0$ is the actual set point at 186.65 hPa (140 Torr) and $\Delta p$ is a pressure difference depending on the water vapor mole
fraction. If $\Delta p$ is small enough it can be assumed that the pressure changes linearly and the peak areas of the absorption lines follow

$$A(p) = A(p_0) * \left(1 + \frac{\Delta p}{p_0}\right). \tag{11}$$

Substituting $A_{CO2}(p_0)$ and $A_{H2O}(p_0)$ according to Eq. (11) in Eq. (8) yields:

$$A_{CO_2}^{wet}(p) = A_{CO_2}^{dry} \left(1 - C_{H_2O}(p_0) * A_{H_2O}(p) + \frac{\Delta p}{p_0}\right). \tag{12}$$

Thus, the bias in the sample cell pressure introduces an error to the calibration constant $C_{H2O}(p_0)$, which is proportional to the relative pressure change $\Delta p / p_0$. Experiments with an additional independent pressure measurement (Reum et al., 2017), as well as analysis of the behavior of the proportional valve, which controls the pressure in the sample cell, show that $\Delta p$ is around 0.7 mbar (0.5 Torr) for a water vapor mole fraction of 3 %. Hence, the pressure bias causes an error of <0.4 % (rel.)





to the $CO_2$ dilution calibration method. Note however that the change in cell pressure with humidity is not fully linear (see Figure 2 in Reum et al., 2017), which could be the reason for the slightly systematic shape in the residuals with lower values around 0.25 % water vapor mole fraction (Fig. 6 b).

In summary it can be said, that an uncertainty at percent or even sub-percent level is achievable for the dilution method. Using a conservative estimate of 1 % (rel.) uncertainty for assessing water vapor from the $CO_2$ dilution experiment presented here, added to the 1.3 % (rel.) uncertainty of the dew point mirror calibration, comparison of the dilution based estimate ($H_2O_{dilution} = (1.029 \pm 0.023)*H_2O_{dewmet2013}$) with the FISH calibration bench ($H_2O_{CalBench} = (0.97 \pm 0.04)*H_2O_{dewmet2013}$)( neglecting the offsets) shows an overlap within their combined uncertainty. Note that this also means that the $H_2O$ calibration via dilution of $CO_2$ is statistically consistent with the classical calibration using dew point or frost point hygrometers. This is a promising result for this experiment, especially when considering that different CRDS instruments were used and the comparisons took place two years apart.

Follow on experiments can achieve better and more reliable results for the water calibration by $CO_2$ dilution, if also low water vapor levels (<300 ppm) are measured, the sample cell pressure is corrected for deviations due to different water vapor levels, optimized spectral models and fitting procedures are applied, and sample air with a $CO_2$ mole fraction in the atmospheric range is used. To determine a calibration factor for the water vapor estimates based on peak height measurements, which is the standard measurement method of the CRDS analyzers at the moment, since it provides better short-term precision than the peak area measurements, the experiment can be simplified. As can be seen in Eq. (8) the water vapor mole fraction ($C_{H2O}*A_{H2O}$) can be calculated, if the dry and wet peak areas of the $CO_2$ absorption line are known. Thus, the measurement of the water vapor peak area can be skipped, which reduces the overall uncertainty. On the other hand, for low water vapor mole fractions (<10 ppm) a wrong pressure reading (as described above) has a higher impact since it effects the wet peak area, but not the dry peak area measurement. By looking at the deviation of the ratio between wet and dry peak area to one the error gets enhanced even more.

Obviously the dilution method can be applied to other species, too, and is not limited to $CO_2$ and water vapor. The same principle can be used for any species measurable by a CRDS analyzer, provided that the corresponding dilution effect is large enough to be detectable with sufficient precision.

### 3.2.4 Summary

Table 1 shows in summary the results of the different calibration experiments. The uncertainties of the coefficients for the FISH calibration bench comparison result from the dew point mirror calibration uncertainty and the uncertainty of the calibration bench. For the $CO_2$ dilution effect it is the dew point mirror calibration uncertainty plus a conservative estimate of the dilution method uncertainty. Note that both offsets, or rather their uncertainties, are likely not reliable.

Based on this experiments the calibration constant of 0.802 ± 0.010 from the dew point mirror comparison in 2013 is recommended for the water vapor measurements of the CRDS instrument.



## 4 Analysis of the flight data and comparison with the reference instruments

During the DENCHAR flight campaign between 23 May and 1 June 2011 four inter-comparison flights with a total flight time of about 14 hours were conducted with a Learjet 35A. Starting from an airbase in Hohn (Germany) the flights covered a region ranging from North Germany and Poland to South Norway, the North and the Baltic Sea, and altitudes up to 12.5 km. Hence, also the lower stratosphere was reached. Two instruments served as reference instruments for water vapor measurements. The first was CR-2, a frost point hygrometer with an accuracy of ±0.1°C dew point (manufacturer data, Buck Research Instruments L.L.C., US, www.hygrometers.com). The second reference instrument was the Fast In-situ Stratospheric Hygrometer (FISH), which is based on the Lyman-α photofragment fluorescence technique and has a total accuracy of 6-8 % in the range from 4 to 1000 ppm and down to 0.3 ppm for lower mixing ratios (Meyer et al., 2015). FISH measured total water instead of water vapor during the campaign, since the inlet system of FISH allowed for sampling of cloud droplets and ice crystals. Both instruments were operated by the research centre Jülich.

### 4.1 Results and Discussion

### 4.1.1 Measurement repeatability

To assess the measurement repeatability of the CRDS analyzers water vapor measurements during the periods with stable atmospheric conditions were selected. Of course, there are still natural variations left in the data, therefore only upper limits of the repeatability can be estimated. After correcting for offset and calibration (according to the 2013 dew point mirror comparison, calibration factor = 0.802 ± 0.010), the standard deviation of the difference between the 0.4 Hz data and the 60 seconds averages is calculated as a measure of short-term fluctuations. In order to avoid additional noise from variations in sample cell pressure, periods with unstable sample cell pressure were neglected. Deviations of the sample cell pressure from its set point of 186.65 hPa can occur during sudden, fast changes in altitude for which the pressure adjustment is too slow to adapt. Figure 7 shows the resulting short-term fluctuations (i.e. the standard deviations of the difference between the 0.4 Hz data and the 60 second averages) for different water vapor ranges. The significance of the results certainly depends highly on the number of data, which were available to calculate the standard deviations in each water vapor interval. Thus, in order to find a reliable estimate for the measurements, results based on a larger number of data are highlighted. Although high scatter of the data between 30 and 100 ppm makes it difficult to find a reliable estimate, the flight data suggest an upper limit for the measurement repeatability of 4 ppm or 5 % (rel.) (whichever is greater) for water vapor <100 ppm, and 5 % (rel.) or 30 ppm (whichever is smaller) for water vapor >100 ppm, indicated by the three black lines in Fig. 7.

For comparison repeatability estimates of the CRDS water vapor measurements determined under laboratory conditions, at 2.5 s time resolution and for an integration time of 30 s, are shown in Table 2. They were derived from experiments during which the CRDS analyzer measured pressurized, dried ambient air from a high-pressure tank, which was humidified by a dew point generator (Li-610 from Li-Cor) to specific water levels. For water vapor <100 ppm the results of the flight and




laboratory data are in good agreement. For water vapor >1000 ppm the laboratory data indicate that a repeatability of 30 ppm for the flight data is a very conservative estimate, which is most likely due to natural variations in the atmosphere.

Compared to the reference instruments the repeatability of the CRDS analyzer is worse at low water vapor levels (<100 ppm), but comparable at higher levels.

### 4.1.2 Response time

Figure 8 and Figure 9 show selected time periods of the third flight on 31 May and the fourth flight on 1 June, respectively. In addition to the offset corrected and calibrated (according to the 2013 dew point mirror comparison, calibration factor = 0.802 ± 0.010) water vapor measurements of the CRDS analyzer, in the following simply referred to as CRDS measured water vapor, shown here as black points (30 seconds mean as grey points) and the flight data of the reference instruments CR-2 (dark blue squares) and FISH (light blue triangles), also water vapor measurements of two additional analyzers, which took part in the inter-comparison campaign, are presented. Flight data of WaSul-Hygro, a tunable diode laser-based dual-channel photoacoustic humidity measuring system (Tátrai et al., 2015), are shown as orange diamonds; flight data of the Selective Extractive Airborne Laser Diode Hygrometer (SEALDH-1), which is based on tunable diode laser absorption spectroscopy (Buchholz et al., 2013), are shown as green triangles (please note: not to be confused with the currently used new instrument SEALDH-II, which has a much better performance and smaller uncertainties). Furthermore, the saturated water vapor (violet points) is added to point out that the measurements were taken outside of clouds.

The flight data of all analyzers in Fig. 8 and Fig. 9 indicate that the response time of the CRDS is not slower than any other instrument. This applies for the whole water vapor range and for both transition directions: from wet to dry conditions as well as from dry to wet. Thus, the low sample gas flow of 100 sccm and the one meter long inlet line cause no disadvantages. As expected, the slowest response is shown by the CR-2, whose measurement signal tends to overshoot and oscillate after fast changes in water vapor.

Results of a simple laboratory test, where a three way valve was used to switch between wet and dry air, allowed to estimate the 10-90 % rise and 90-10 % fall times as 6-7 seconds and the recovery time to 99 % of a challenge as 25 seconds. For a step from 2.3 % (around 20°C dew point) to 10 ppm water vapor mole fraction the measurement takes about 200 seconds to get down. The times are pretty much identical wether or not the 50 cm long inlet line is included.

### 4.1.3 Comparison to reference instruments

Figure 10 shows the in-flight CRDS measured water vapor (in black) and the reference instruments CR-2 (dark blue) and FISH (light blue), as well as the corresponding atmospheric pressure levels (in green), for all four flights. For better comparison also the 30 seconds mean of the CRDS data is shown (in grey). Due to an internal leak FISH could not deliver reliable data for the first two flights. CRDS data of the first flight after around 1 pm were influenced by icing of the inlet, since the deicing of the Rosemount inlet was accidentally not switched on during that flight.



The absolute water vapor differences between the three instruments for different water vapor intervals can be seen in Fig. 11 (CRDS - CR-2 as black points, CRDS - FISH as dark blue diamonds, CR-2 - FISH as light blue triangles). The water vapor measurements of CR-2 are chosen as x-axis, because they cover all flights in contrast to the FISH data. The differences are calculated from the 30 seconds mean data of all analyzers and are averaged over intervals of 1 ppm, 10 ppm, 100 ppm, 1000

ppm, and 10000 ppm water vapor in the corresponding water vapor ranges of 0-10 ppm, 10-100 ppm, 100-1000 ppm, 1000-10000 ppm, and >10000 ppm, respectively. The standard deviations of the averaged differences are shown as error bars. For plotting reasons all differences <1 ppm are set to 1 ppm. CRDS data influenced by icing during the first flight are neglected. Likewise, measurement data of all instruments in the presence of clouds are excluded, since FISH measured total water. Based on observations during the flights, which are recorded in the flight logs, this concerns in particular all measurements

made between 11:13 am and 11:40 am on flight four (1 June).

A reliable evaluation is hard to make as the reference instrument FISH sampled data only for two flights and for flight 4 the measurements diverge significantly from the CR-2 data to a large extent. Moreover, the slow response of the CR-2 and the oscillations of the signal after sudden changes in water vapor are problematic for the comparison. Furthermore, flight data between 11:13 am and 11:40 am for flight 4 could not be used, due to the occurrence of clouds. However, it is interesting

that also the CRDS water vapor measurements deviate from the CR-2 during that period, as can be seen in Fig. 10. Figure 12 shows a closer look at this cloud-affected flight section. The CRDS measured water vapor (black points, 30 seconds mean as grey points) are plotted together with flight data of CR-2 (dark blue squares), FISH (light blue triangles), Wasul-Hygro (orange diamonds) and SEALDH-I (green triangles). The latter two show approximately the same behavior as the CR-2, while the CRDS shows a behavior similar to that of FISH (measuring total water) with $H_2O$ mole fractions within clouds

larger than that corresponding to saturated water vapor (violet points). This might indicate that also the CRDS samples cloud particles, i.e. the separation in the Rosemount air inlet of ice particles and water droplets from the sample air is not fully efficient. In fact relative humidity measurements of the MOZAIC humidity device, which uses the same type of Rosemount Inlet housing, occasionally show similar artifacts, when measuring within clouds, that contain liquid water particles (air temperature > -40°C) (Smit et al., 2013). Most likely some small ice particles and water droplets are able to follow the sharp

right angle turn of the minor air flow into the inner part of the Rosemount housing, instead of flying straight through the main channel of the housing (see Figure 2.6 in Smit et al., 2013). However, due to the very short time period the sample air stays inside the housing until it passes the sensor elements and leaves again through a small outlet, only the liquid water droplets can evaporate fast enough to be observed by the humidity device. In contrast, as can be seen in Fig. 12, the CRDS measurements do show cloud artifacts also at air temperatures (black line) below -40°C, i.e. in pure ice clouds. Most likely

the reason for this is that water droplets and ice particles enter the inlet line of the CRDS and are evaporated within the inlet line or at the heated inlet filter of the CRDS. Meaningful statistics about how often droplets and ice particles are measured in clouds can be obtained as soon as more flight data of the CRDS analyzer are available within the IAGOS project, since every IAGOS aircraft equipped with the GHG-package is also equipped with the MOZAIC humidity device and a cloud probe.





The absolute differences in Fig. 11 indicate a positive difference between the CRDS and CR-2 of <10 % or 10 ppm (whichever is greater) for water vapor ranges >10 ppm. FISH has a negative deviation to both instruments in that range. For water vapor >100 ppm the data imply a difference of 10-20 %. For the interval of 10-100 ppm water vapor the difference to the CRDS is around 10 %, to CR-2 about 10 ppm. At very low water vapor (<10 ppm) the reference instruments show a

good agreement during flight 3, but disagree strongly during flight 4. On average the CR-2 has a positive bias <2 ppm against FISH. For the CRDS the water vapor data suggest a positive bias <2-3 ppm to the CR-2, but the measurements are highly affected by the slow response of the CR-2. Comparison to FISH likewise indicates a positive difference <2-3 ppm. During comparison against the FISH calibration bench the CRDS analyzer showed a positive bias of 12.2 ppm (see Sect. 3.2), which strengthens the presumption that the air hasn't been in equilibrium for the lowest water vapor level measured

during the experiment.

Meyer et al. (2015) report an agreement of FISH with other in situ and remote sensing hygrometers under field conditions of about ±5–20% @ <10 ppm and ±0–15% @ >10 ppm. Thus, results of the comparison between CRDS and FISH during the DENCHAR inter-comparison campaign are at the upper end of that range.

**Conclusions**

During the DENCHAR inter-comparison flight campaign in Hohn (Germany) in May-June 2011 a commercial cavity ring-down spectroscopy (CRDS) based gas analyzer (G2401-m, Picarro Inc.,US) was installed on a Learjet to measure atmospheric water vapor, $CO_2$, $CH_4$ and CO. The components of the instrument and the inlet system are identical to those chosen for the IAGOS-core Greenhouse Gas package.

For the calibration of the water vapor measurements three different methods were tested. The standard calibration of the

CRDS analyzer is the comparison against a dew point mirror (Dewmet TDH, Cooled Mirror Dewpointmeter, Michell Instruments Ltd., UK) in the range from about 0.8 % to 3.0 % water vapor mole fraction. If the dew point mirror is calibrated regularly by the manufacturer, the accuracy of this calibration method is limited by the uncertainty range of the dew point mirror (1.3 % (rel.)). A comparison against the FISH calibration bench, during the DENCHAR flight campaign, in the range from 2-600 ppm water vapor, confirmed that the extrapolation of the dew point mirror calibration down to low water vapor

levels is possible, and that the standard calibration of the CRDS analyzer is in agreement with the FISH calibration within the 4 % uncertainty range of the FISH calibration bench. Furthermore, a new and completely independent calibration method, which is based on measurement of the dilution effect of water vapor on the $CO_2$ mole fraction, was presented. This new method was found to agree with the dew point mirror calibration within 2.9 % (rel.) in the water vapor range from 300 ppm to 2.7 %. Assuming a conservative 1 % (rel.) uncertainty for the $CO_2$ dilution method, comparison of the dilution based

estimate with the FISH calibration bench showed an overlap within their combined uncertainty. Thus, the water vapor calibration via dilution of $CO_2$ is statistically consistent with the classical calibration using dew point or frost point





hygrometers. The dilution method can be used for other species, too, provided they are measurable by a CRDS analyzer and the corresponding dilution effect is large enough to be within the detection limits.

An upper limit of the precision of the water vapor measurements by the CRDS was determined from flight data of the DENCHAR inter-comparison campaign, as 4 ppm or 5 % (rel.) (whichever is greater) for water vapor <100 ppm, and 5 %
(rel.) or 30 ppm (whichever is smaller) for water vapor >100 ppm. A more reliable estimate will be possible as soon as more $H_2O$ flight data are available. During the four DENCHAR flights the CRDS analyzer showed a good time response (10-90 % rise and 90-10 % fall times: 6-7 s, recovery time to 99 % of a challenge: 25 s) and long-term stability for the water vapor measurements. Comparison against the reference instruments was difficult, due to lack of data availability of FISH, the slow response of CR-2, the exclusion of data, which were affected by clouds, and the partly poor agreement between FISH and
CR-2. However, for water vapor levels >10 ppm the flight data imply a negative difference between the CRDS and FISH from about 10-20 % and a positive difference between the CRDS and CR-2 of <10 % or 10 ppm (whichever is greater). For water vapor <10 ppm the flight data suggest a positive bias of <2-3 ppm to both, FISH and CR-2.

Accuracy of the CRDS instrument was estimated, based on the laboratory calibrations, as 1 % (relative) for the water vapor range from 2.5 % down to 0.7 %, than increasing to 5 % (relative) at 50 ppm water vapor. Accuracy at water vapor mole
fractions below 50 ppm was difficult to assess, as the reference systems suffered from lack of data availability.

Future deployment of the CRDS system within IAGOS will help to further evaluate the performance, via better statistics and long-term comparison to the MOZAIC humidity device, which is deployed on each IAGOS aircraft. Thus, essential water vapor measurements, including regular in situ data in the sensible UTLS region and vertical profiles of $H_2O$ in the troposphere and lower stratosphere for major parts of the globe, are expected to be delivered for validation of remote sensing
based observations from satellites and ground, for the improvement of the performance of climate models and weather forecasts, or for climate trend studies.

**Competing interests**

The authors declare that they have no conflict of interest.

**Acknowledgements**

This work was supported by the European Commission through the FP6 project IAGOS Design Study (contract number 011902-DS) and IAGOS-ERI, an FP7 project (grant agreement no. 212128). The DENCHAR flight campaign was funded by the European Commission's FP7 project EUFAR. Funding for IAGOS Deutschland was provided by the Bundesministerium für Bildung und Forschung (BMBF), Germany. Additional funding from the German Max Planck Society to support the instrument development is greatly acknowledged. Development and operation of Wasul-Hygro was supported by the
Hungarian Research and Technology Innovation Fund (OTKA), project no. NN109679.



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



**Table 1: Overview of the different calibration methods.**

| method | water vapor mole fraction range | result |
|---|---|---|
| dew point mirror (Dewmet) from 2013 | 0.7–2.5 % | $H_2O_{Dewmet2013} = (0.802 \pm 0.010) * H_2O_{uncalibrated}$ |
| FISH calibration bench | 2–600 ppm | $H_2O_{calBench} = (0.97 \pm 0.04) * H_2O_{Dewmet2013} - (12.2 \pm 0.5)$ ppm |
| $CO_2$ dilution effect | 300 ppm - 2.7 % | $H_2O_{dilution} = (1.029 \pm 0.023) * H_2O_{Dewmet2013} - (16.1 \pm 1.6)$ ppm |





**Table 2: Reapeatability estimates of the CRDS water vapor measurements derived from laboratory experiments.**

| water vapor mole fraction [ppm] | repeatability at 2.5 s time resolution [ppm] | repeatability at 30 s integration time [ppm] |
|---|---|---|
| 3 | <6 | <2 |
| 30 | <10 | <5 |
| 5000 | <9 | <5 |
| 8000 | <10 | <2 |
| 12000 | <10 | <4 |
| 19000 | <12 | <6 |





**Figure 1: Cross section of the inlet line (green) mounted into a Rosemount Total Air Temperature housing (model 102B, adapted from Stickney et al. (1994)). The inlet line is pointed orthogonal to the airflow through the housing to prevent from sampling larger aerosols, ice particles, and water droplets.**



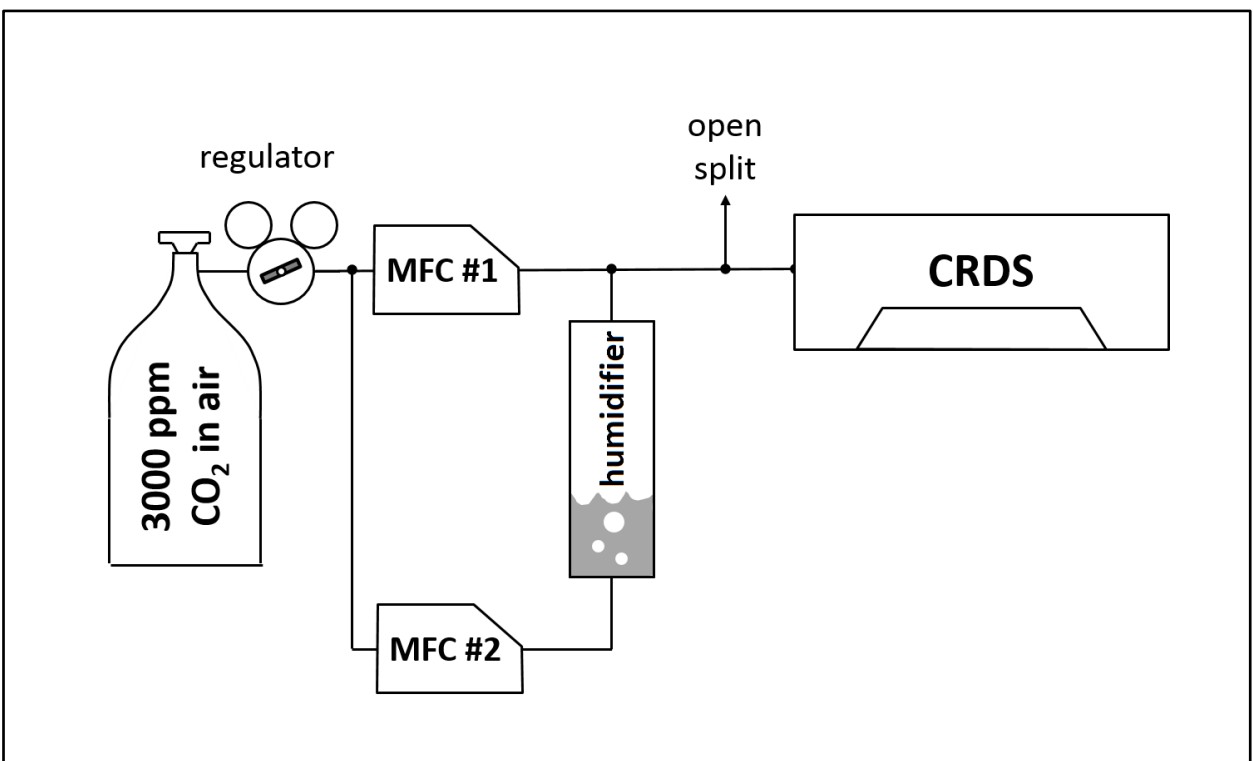

**Figure 2: Experimental setup for the water vapor calibration by the CO₂ dilution effect.**





**Figure 3: a) Uncalibrated water vapor measurements of the Picarro CRDS analyzer (CFKBDS2003) against the measurements of the dew point mirror (Dewmet TDH) during the 2013 calibration. The corresponding fit is shown as blue line, the calibration curve of the 2009 comparison as a grey line. The uncertainty of the fitted slope is composed of the fitting error and the uncertainty of the dew point mirror. b) Water vapor difference between the 2009 and 2013 experiments. The error bars indicate the uncertainty range, which results from the combination of the dew point mirror uncertainties during the 2009 and the 2013 calibration.**



**Figure 4: a) Uncalibrated water vapor mole fractions of the two Picarro CRDS analyzers CFADS37 and CFKBDS2003 during a comparison experiment in 2014 (black points). The blue line indicates the quadratic fit of an earlier comparison between the same instruments in 2011. The differences between the two comparisons, thus the drift of the two analyzers over the three years from 2011 to 2014, are shown in plot (b).**







**Figure 5: a) Dry air water vapor mole fractions measured by the CFKB2004 CRDS analyzer and the FISH calibration bench during a comparison in 2011. The CRDS data are calibrated according to the 2013 comparison against a dew point mirror. The corresponding fit is shown in blue. Residuals (difference between water vapor mole fractions measured by FISH calibration bench and the linear fit) can be seen in the plot (b). Error bars indicate the uncertainty range of the calibration bench of 4 % (rel.).**





**Figure 6: a) Water vapor mole fraction based on the $CO_2$ dilution method plotted against the water vapor mole fraction measurement of the CRDS analyzer (offset corrected and calibrated according to the comparison against a dew point mirror in 2013) during the fine scan experiment. The corresponding fit is shown in blue. Residuals (difference between water vapor mole fraction based on dilution method and the linear fit) can be seen in plot (b).**





**Figure 7: Standard deviation of the difference between the 0.4 Hz data (offset corrected and calibrated according to the 2013 dew point mirror comparison) and the 60 second averages, averaged for intervals of 1 ppm, 5 ppm, 10 ppm, and 100 ppm water vapor in the corresponding water vapor ranges of 0-10 ppm, 10-100 ppm, 100-1000 ppm, 1000-10000 ppm, respectively. Different colors and symbols indicate different flights. Results with higher priority are highlighted. The horizontal and diagonal black lines indicate standard deviations of 4 ppm, 30 ppm, and 5 % (rel.) respectively.**





**Figure 8: Water vapor mole fractions of the CRDS analyzer (offset corrected and calibrated according to the 2013 dew point mirror comparison) (black points, 30 seconds mean as grey points), the CR-2 (dark blue squares) and FISH (light blue triangles) instruments, as well as the WaSul-Hygro (orange diamonds) and SEALDH-I (green triangles) analyzers, for a time period during the flight on 31 May 2011.**







Figure 9: Same as Fig. 8, but for a time period during the flight on 1 June 2011.





**Figure 10: In-flight water vapor data of the CRDS analyzer (in black, 30 seconds mean in grey) and the reference instruments CR-2 (in dark blue) and FISH (in light blue) for the four flights on 26 May, 30 May, 31 May, and 1 June 2011. The corresponding atmospheric pressure levels are shown in green. The CRDS data are offset corrected and calibrated according to the 2013 dew point mirror comparison.**



**Figure 11: Absolute differences of the 30 seconds mean CRDS and CR-2 in-flight data (black points), CRDS and FISH data (dark blue diamonds), and CR-2 and FISH data (light blue triangles) averaged over intervals of 1 ppm, 10 ppm, 100 ppm, 1000 ppm, and 10000 ppm water vapor in the corresponding water vapor ranges of 0-10 ppm, 10-100 ppm, 100-1000 ppm, 1000-10000 ppm, and >10000 ppm, respectively, against the CR-2 water vapor flight data. Error bars indicate the standard deviations of the average differences.**



**Figure 12: Water vapor mole fractions of the CRDS analyzer (offset corrected and calibrated according to the 2013 dew point mirror comparison) (black points, 30 seconds mean as grey points), the CR-2 (dark blue squares) and FISH (light blue triangles) instruments, as well as the WaSul-Hygro (orange diamonds) and SEALDH-I (green triangles) analyzers, during the flight on 1 June 2011, in the presence of clouds. Also shown are the water vapor mole fraction corresponding to saturation (violet points) and the static air temperature (black line).**