# Peer review of "Evaluation of the IAGOS-Core GHG Package H2O measurements during the DENCHAR airborne inter-comparison campaign in 2011"

_Atmospheric Measurement Techniques, 2018_

## Referee Comment (RC1) · Anonymous Referee #1 · 18 Apr 2018

This manuscript describes the new water vapor instrument for IAGOS commercial flight measurements, its calibration (in several ways), and some preliminary comparisons with other water vapor instruments during an aircraft-based inter-comparison campaign in 2011. The paper is, in many instances, very detailed and provides ample descriptions of the calibration and evaluation, but at times does not provide the information necessary to completely understand some of the findings. My finding is that the paper can be published in AMT if the shortcomings (noted below) are improved.

General Comments

I don't agree with the measurement "repeatability" values determined in this paper

for WV mixing ratios < 100 ppm. On page 14 it is stated that an "upper limit for the measurement repeatability" ... is ... "4 ppm or 5%, whichever is greater for WV < 100 ppm". If that is the case, why are there so many data points for WV < 100 ppm in Figure 7 that have standard deviations > 5 ppm (5%)? The upper limit estimate for WV > 100 ppm is much more in line with the Figure 7 data.

The abstract is very, very detailed, and can be shortened by describing the overall work in more general terms. An example of too much detail is the inlet system description that is described again in similar (or even less) detail on page 6. Also, what were the quantitative goals of comparing the CRDS instrument to others? In other words, what levels of performance were you hoping to obtain from the CRDS in terms of precision, accuracy and stability?

The frequent interchange between using ppm and % as units of mixing ratios (mole fractions) can lead to confusion, especially when it is desired to report relative differences or uncertainties (in percent). An example is Lines 5-6 on page 10 where "relative" is required to differentiate relative differences (in %) from mole fractions (in %). Mentally converting % to ppm is not terribly difficult, but by changing all the text and graph axes from % to ppm, you would completely avoid any potential confusion. Some of your 2-panel figures show mixing ratios in % (top panel) while differences or residuals are presented in ppm (bottom panel).

In many places uncertainties and error bars are presented without any indication if they are based on 1 or 2 standard deviations of the mean. It is helpful to report uncertainties, but only if the reader knows on which statistics they are based.

Please refrain from including Figure caption information in the main body text of the paper. It makes the paper more tedious to read. This occurs in numerous places throughout the manuscript.

During the comparison flights, did all instruments sample the airstream from a common Rosemount TAT inlet, or did each have its own inlet? I believe only the FISH inlet is

described as being different. If there was a common inlet for the other instruments, how was the sample stream physically split between them? If each instrument had its own inlet, how much of a difference might the different types and/or locations of inlets play in the overall comparisons? The absence of information about the sampling inlet for each instrument makes it very difficult to understand the inlet-dependent discussion on Page 16. I therefore suggest that this discussion be omitted from the paper because (a) it requires adding much greater detail about inlets and (b) inlet influences aren't necessarily a part of the evaluation of the CRDS measurement capabilities for IAGOS.

The clarity of the presentation within this manuscript would be significantly improved through editing by a native English speaker. Hopefully that can be done before revisions are submitted.

Specific Comments:

Page 1, Line 16: "North-Germany" is not a proper noun, so "North" should not be capitalized and the hyphen should be omitted, i.e., "northern Germany". This occurs several times within the manuscript, e.g., Pg 4 L19.

Pg 1, L19: What (specifically) is meant by "assessment validation"? Are there targets for the measurement precision and/or accuracy of this new IAGOS instrument that are the basis for its evaluation as described in this paper?

Pg 3, L6: "data of sufficient quality in the UTLS" - sufficient for what? This comment ties in with my previous comment about targets for "sufficient" measurement precision and/or accuracy.

Pg 3, L12-17: why include this long list of satellite-based water vapor sensors when their data are not included in this paper?

Pg 3, L18-19: "insufficient spatial resolution" - what spatial resolution is sufficient? And don't you instead mean "vertical resolution" here when discussing satellite and remote-sensing observations?

Pg 4, L6: "The central problematic of all these different ... are the remaining, unexplained discrepancies ...". Are the discrepancies "remaining"? From what? Are they a problem? It depends on the science you are investigating. Please improve the clarity of this statement.

Pg 4, L15-18: This is the type of general statement that belongs in the abstract instead of the very detailed descriptions.

Pg 4: I find a lot of the information presented in the abstract is duplicated here in the introduction. The introduction should explain why your investigation is important and how you intend to perform it. Specific information about instruments, including the MOZAIC humidity device, belongs in a separate section describing the various instruments.

Pg 5: There is a lot of repeated information on this page. Please ensure that you don't write the same sentence more than once in the paper, with the exception of abstract and conclusions.

Pg 5, L24: What "impact on gas density and spectroscopy" is being minimized? This statement is vague and needs clarification.

Pg 6, L4-6: This description was already given in great detail, in the abstract (where it likely does not belong).

Pg 6, L13: How are there only small humidity differences between the cabin and the outside air?

P6 6, L27-30: This would be the best place to explain that the instrument zero ("offset") is not stable (over what time scale?) and requires frequent adjustment, while the instrument gain is very stable and needs calibration only infrequently.

Pg 9, L30-32 and Figure 3: Are the reported uncertainties 1-sigma? 2-sigma? This comment applies to the entire paper, wherever uncertainties are given (text and Figures).

Pg 10, L1: "0.0024 ± 0.0021" is "not significant"? What does "significant" mean here? It can't possibly mean "not statistically different from zero" as the reader might expect.

Pg 10, L12-14, L20: The differences between calibration curves presented in ppm in Figures 3b and 4b and expressed in ppm and % (rel) in the text may cause confusion here. If you express all mixing ratios, differences and residuals in ppm it avoids any confusion. For the Figures 3b and 4b it would add value to include a right axis of relative differences/residuals. Given the near-linearity of the differences/residuals against mixing ratios, I think the relative data (different symbol/color) would be quite constant over the entire range of mixing ratios.

Pg 10, L26-28: Do you have enough evidence to make this statement based on a conjecture that the instrument "was not calibrated well enough" in 2009? This seems like a hand-waving statement with little factual backing. Is there further evidence that allows you to select the 2013 calibration as being correct?

Pg 10, L33: I have to assume that the mixing ratio ranges compared with each other instrument were chosen based on the best measurement ranges of the other instruments. Is this the case? If so, please inform the reader of this earlier in the paper. I see the range information in Table 1, but the table is not mentioned until page 13. An earlier statement about the choices of comparison ranges for different instruments and a mention of Table 1 are needed, otherwise it appears the comparison ranges were chosen randomly.

Pg 11, L5-13 and Figure 5: The residuals in Figure 5b (presented in ppm) are discussed in the text in relative (%) terms. Adding a right axis for relative residuals to Figure 5b (using different symbol/color) would strengthen the discussion of relative residuals in the text.

Pg 11, L30-31: It is odd that the residuals at ∼250 ppm are negative (Figure 6b) while at the lowest mixing ratio they are positive. With an offset of 16 ppm one would expect the residuals at both of the lowest mixing ratios to be of the same sign. Any idea why

they are negative at $\sim$250 ppm and positive at the lowest mixing ratio?

Pg 12, L1: If "the lowest measurement was made at around 300 ppm", why are there data points below 300 ppm in Figure 6?

Pg 12, L6-7: "which was tried to exclude" is an example of where a native English speaker could help clean up the grammar in this manuscript.

Pg 12, L10: It's not "zero air" if it contains 2 ppm CH4. You might want to use a different term.

Pg 12, L20: Of course, "the pressure sensor is sensitive to water vapor", as it is to any gas-phase constituent. I think you are trying to convey that the pressure sensor responds in a non-linear way to increasing amounts of water vapor, creating a water vapor-dependent bias in the pressure readings. In L25, what does "assumed that the pressure changes linearly" mean here? If the pressure changes linearly with changes in water vapor you don't have a problem with water vapor-dependent biases. I don't understand this assumption.

Pg 12, L12-25: Don't the water vapor-dependent bias in the pressure readings also affect the in-flight data of the CRDS? Here you dwell on how this affects the CO2 dilution calibration method, but don't discuss the effects on the actual in-flight measurements. If this also affects the flight data, why only discuss it here in the "Calibration by CO2 Dilution" section?

Pg 13, L26: It is strange to have a section entitled "Summary" in the middle of this paper. How about "Calibration Summary" instead?

Pg 13, L30: "Note that both offsets, or rather their uncertainties, are likely not reliable." So, are the "offsets" (y-intercepts) likely not reliable, or their uncertainties, or both?

Pg 14, L8-9: What is "a total accuracy" and what does "and down to 0.3 ppm for the lower mixing ratios" mean? How low is "lower"?

[Figure]

Pg 14, L14-15: What are the "stable atmospheric conditions" mentioned here? This sentence needs a comma after "analysers".

Pg 14, L13: Why use the term "repeatability" instead of precision? From the subsequent description it appears that you are looking at variations in the 0.4 Hz measurements from the 60-second averages. I would call this "precision" and not repeatability, since repeatability can imply stability in results obtained at different times, such as re-sampling the same air mass five minutes later and seeing how "repeatable" the measurements are.

Pg 14, L18-19: Isn't the 60-second average calculated from the 2.5-second measurements? Then why do you calculate the standard deviation of their differences when the standard deviation of the 60-second average directly provides a direct statistical measure of variability in the shorter-term measurements?

Pg 14, L24: "results based on a larger number of data are highlighted". Aren't most 60-second averages based on 24 measurements made at 0.4 Hz?

Pg 14, L26-27: "upper limit" ... "4 ppm or 5%, whichever is greater for WV < 100 ppm". In Figure 7, why are there so many data points for WV < 80 ppm that have standard deviations > 4 ppm (5%)? The upper limit estimate for WV > 100 ppm is much more in line with the Figure 7 data: most of the data points for WV > 100 ppm lie below the 30 ppm standard deviation line and to the right of the 5% diagonal. This cannot be said about the WV < 100 ppm data, many of which lie above the 4 ppm standard deviation line and to the left of the 5% diagonal. Please explain the connection between this Figure and the numbers you have extracted from it that appear to underestimate the precision values for measurements at WV < 100 ppm.

Pg 15, L9-16: No need to describe the different symbols since that is done in both the Figure 8 caption and legend. What is more important here is a description of what the Figure shows about response time of the CRDS. I think that showing only a short period of Figure 9 would better support this discussion of response times <10 seconds.

Specifically, the flight segment from 12.35 to 12.55 UTC on 1 June 2011 when water vapor increases and falls an order of magnitude in about 3 minutes.

Pg 15, L22-25: This discussion of laboratory tests of response time is not very informative or conclusive because the reader has no idea how much water vapor is in the "wet and dry air". Also, "recovery time to 99% of a challenge" is not understandable.

Pg 15, L27-31 and Pg 16 L1-6: Please remove Figure 11 symbol descriptions and the explanation of data binning since all this information appears in the Figure caption.

Pg 16, L1-2 and Figure 11 caption: "Absolute" differences steers my thinking towards absolute values, which are definitely not what Figure 10 shows. "Absolute" is unnecessary in both instances because the differences are clearly in units of ppm, i.e., not relative differences.

Pg 16, L4-6 and Figure 11: If the differences are binned by CR-2 water vapor in 1 ppm bins for CR-2 < 10 ppm, why are there multiple blue diamonds and multiple blue triangles in the lowest CR-2 mixing ratio bin (and some other bins)? Aren't the averages computed from all segments of all flights? If not, why are the average differences in each bin determined for subsets of the 4 flights?

Figure 11: The log-scale Y-axis is problematic for average differences < 1 ppm and for the error bars that appear asymmetric around the mean. Why don't you instead plot average relative (%) differences using a linear-scale Y axis? Average differences < 1 ppm will then be displayed correctly and the error bars will be symmetric around the mean values. I think the relative values will be much more like flat lines with CRDS-CR2 values above zero and the other instrument pairs below zero.

Pg 16, L11-33: This section is difficult to understand because there are no descriptions of the various inlets for the different instruments. All that is known at this point is that the FISH inlet is different from the CRDS inlet, although there is now a hint that the other instruments do not share the CRDS inlet. Is it really important or necessary to

include this section of the paper when it requires adding more details of the various instrument inlets? This section seems to cover a different topic from the main thrust of this paper and may even be worthy of a separate paper in the future.

---

## Referee Comment (RC2) · Anonymous Referee #2 · 9 May 2018

This manuscript describes the calibration and intercomparison of a Picarro CRDS instrument for measurement of ambient water vapor. Multiple laboratory calibrations of the instrument by comparison with a NIST-traceable dewpoint hygrometer, the FZ Jülich calibration setup, and a new method using co-measured CO2 dilution by the added H2O are described to assess the instrument accuracy and stability over time (up to years). Comparison of in-flight measurements with several research hygrometers are presented to assess the instrument performance. The CRDS instrument described is the model that has been selected for the IAGOS-Core payload for regular in-service measurements, and therefore the assessment provided will have bearing on a significant atmospheric data set. The manuscript topic and material presented are

appropriate for publication in AMT. The manuscript would benefit from some changes to the analysis, structure and presentation.

General Comments:

The manuscript would benefit substantially from copyediting to correct usage and punctuation. Examples include: the use of commas after introductory prepositional phrases,

The inclusion of water vapor mixing ratios in both ppm (ppmv) and % units is somewhat awkward, although I understand that those units are regularly used for water in different environments. Suggest consistently using ppmv for the manuscript with an initial parenthetical equivalence for the reader, e.g. on line 30 "7000 ppmv (0.7%) to 25,000 ppmv (2.5%)". This would have the additional benefit of simplifying the discussion of the uncertainty in % of measurement without needing to specify "(rel.)". The DENCHAR website uses ppmv even for high values mentioned.

The details of representations (e.g. blue line, dark blue triangle) of data in the plots should not be included in the text, only in the figure captions (where they should be fully described for clarity). The text should discuss the interpretation and implications of the data presented in the figures. The figure captions could be more descriptive.

The term "repeatability" is used in a number of places in the manuscript (e.g. section 4.1.1) that would appear to be better described as measurement precision. Repeatability is typically the measure of how consistently an instrument will produce the same response when challenged at different times with the same input—related to drift. Precision is more typically used for the short time variation of measurement (as is the case here for the variation about the 60 s or 30 s mean value) or repeated consecutive measurements of a constant input.

Some pieces of information are repeated numerous times within the manuscript out of a desire to be clear, but it seems to go too far. For example, it should be stated at the first reasonable opportunity in the manuscript that 0.802 calibration factor from 2013
will be used going forward and then it need not be repeated every time.

Specific Comments:

P1, L16: "North-Germany" should be "northern Germany"; similarly elsewhere, "South-Norway" and "North-Poland" should be "southern Norway" and "northern Poland"

P1, L23: This paragraph seems inappropriate for the abstract–suggest removing and including the details in the instrument description section.

P1, L30: "instruments" should be capitalized.

P1, L31+: "on the ground over the range from 2 to 600 ppmv against the dew point hygrometer used for calibration of the FISH instrument. A new, independent calibration method based on the dilution effect of water vapor on CO2 was evaluated."

P2, L3: "than" should be "then", but even better could just be deleted–"1% for the water vapor range from 25,000 ppmv down to 7000 ppmv, increasing to 5% at 50 ppm"

P3, L6: sufficient quality for what?

P3, L7: "difficulty representing"; overall this sentence is awkward and difficult to parse.

P3, L18: for what purpose do the satellite measurements have insufficient spatial resolution? Horizontally? Vertically?

P3, L32: What is meant by "beneath"?

P5, L28: "independent on ambient, respectively cabin pressure" is confusing. "independent of both ambient and cabin pressure"?

P5, L29: the filters in the sample line are likely contributing significantly to the long tail in the time constant.

P7, L19: the use of $\pm 125$ ppm as a metric for stability over a range that includes 3 ppm does not induce confidence–would be better to use a relative measure.

P8, L15: "the permeation is negligible" for 3500 sccm is technically incorrect since the permeation rate through the tubing is largely flow independent. The contribution of the permeation to the water vapor mixing ratio in the flow might be argued to be negligible.

P9, L5: "gets only changed by" → "is only affected by"

P9, L13: "whereby it required about" → "requiring"

P9, L29: it would be helpful to the reader to convert the °C uncertainty of the Dewmet to mixing ratio (using the relevant operating pressure) for comparison with other uncertainties.

P11, L11: the CRDS and FISH cal system can't agree within 3% at mixing ratios below 400 ppm without first correcting for the 12.2 ppm offset. It seems reasonable that there was either an outgassing or small leak (it doesn't take much) that led to the offset, but that should be argued and removed before stating the agreement in the response (slope).

P12, L3: "likely unrealistic"–should be more descriptive; "the calculated error of 1.6 ppmv for the offset is likely a significant underestimate."

P12, L20: The section is unclear–are you saying that the pressure sensor response to H2O is different from that to other molecules, leading to an incorrect measurement of the wet flow cell pressure that depends on the H2O content?

P13, L4: It would be better to state a best estimate of the CO2 dilution method uncertainty rather than saying that a value "is achievable". What do you estimate that you actually achieved, and is that consistent with the comparisons?

P14, L2: "Hohn, Germany, the flights"

P14, L5: "Hence, also the lower stratosphere was reached." is an awkward construction, this could be better included within the previous sentence.

P14, L10: "total water instead of only water vapor since the forward-oriented inlet resulted in sampling of cloud droplets and ice crystals when present."

The sentence "Both instruments were…" seems unnecessary.

P14, L14: "CRDS analyzer, water vapor measurements during periods with stable"

P14, L26: the value of 4 ppm/5% stated for the <100 ppmv precision seems to be at odds with the plotted standard deviations in the 10-100 ppmv range, which are typically higher than both.

P15, L9: "and the flight data…" → "along with the flight data from the reference instruments CR-2 and FISH. Water vapor measurements from two additional analyzers that participated in the intercomparison campaign are also shown."

P15, L17: "not slower than any other instrument." → "is similar to that of the other instruments."

P15, L30: "CRDS data from the first flight" and "influenced" → "compromised"

P16, L1: "can be seen" → "are shown" or "are presented" or "are plotted"

P16, L7: "are neglected." → "are omitted."

P16, L11: "sampled data only for two flights" → "operated successfully for only two of the four flights"

P16, L20: "This might indicate that also the CRDS samples cloud" → "This indicates that the CRDS sampling was likely also affected by cloud particles"

Looking at the Rosemount inlet, there is a good likelihood that impaction of droplets or ice crystals (shattering) will produce particles small enough to be entrained in the sample flow and ingested into the instrument. The ice crystal fragments would then be small enough to sublimate in the inlet and affect the measurement.

P18, L1: It is unclear exactly what is meant by the statement "The dilution method can be used for other species…".

The dilution method could be used to determined H2O using methane instead of CO2, or are you trying to say that methane could be measured by the method by comparing to CO2?

---

## Author Comment (AC1) · 21 Jul 2018

Author's response to Interactive comment by Anonymous Referee #1 on "Evaluation of the IAGOS-Core GHG Package H2O measurements during the DENCHAR airborne inter-comparison campaign in 2011" by Annette Filges et al.

A version of the manuscript with all changes is added as supplement.

General Comments 1) I don't agree with the measurement "repeatability" values determined in this paper for WV mixing ratios <100 ppm. On page 14 it is stated that an "upper limit for the measurement repeatability" .. is : : : "4 ppm or 5%, whichever is

greater for WV < 100 ppm". If that is the case, why are there so many data points for WV < 100 ppm in Figure 7 that have standard deviations > 5 ppm (5%)? The upper limit estimate for WV > 100 ppm is much more in line with the Figure 7 data.

Measurement repeatability values were revised. The new estimates are 4 ppm for H2O <10 ppm, 20 % or 10 ppm (whichever is smaller) for 10 ppm < H2O <100 ppm, and 5 % or 30 ppm (whichever is smaller) for H2O >100 ppm

Changed/Added in Abstract, page 2 lines 1-2, in Section 4.1.1, page 14 lines 26-27, and in

Conclusions, page 18 lines 4-5: ". . . of 4 ppm for H2O <10 ppm, 20 % or 10 ppm (whichever is smaller) for 10 ppm < H2O <100 ppm, and 5 % or 30 ppm (whichever is smaller) for H2O >100 ppm."

Added in Figure 7: line indicating a standard deviation of 20 %

Added in caption of Figure 7, line 6: ". . . of 4 and 30 ppm, and 5 and 20 % respectively."

2) The abstract is very, very detailed, and can be shortened by describing the overall work in more general terms. An example of too much detail is the inlet system description that is described again in similar (or even less) detail on page 6. Also, what were the quantitative goals of comparing the CRDS instrument to others? In other words, what levels of performance were you hoping to obtain from the CRDS in terms of precision, accuracy and stability?

Lines 23-28 in the Abstract ("The inlet system, . . .. within IAGOS") were deleted.

Added in Section 2, page 6 line 11: "operation of the instrument with a controlled pressure in the sample cell of 186.65 hPa (140 Torr) throughout the aircraft altitude operating range. . ."

Precision at low mixing ratios is known from previous studies to be around 6 ppm for a 2.3 second integration time, but is related to white noise, such that the corresponding

precision for an integration time of 2300 seconds drops to below 0.3 ppm (Filges et al, 2015). Regarding stability, there is not much known for the CRDS system, also regarding accuracy. For CO2 measurements the system has shown excellent stability, but it is unclear if this is the case for water vapor. So rather than having clear targets, the aim was to see the results from the comparison against reference instruments, and then limit the use of the data to those areas where the quality is sufficient.

Added in Abstract, page 1 line 19: "... project. Since the quantitative capabilities of the CRDS water vapor measurements were never evaluated and reviewed in detail in a publication before the ..."

Added and changed in Section 1, page 4 lines 28-29: "In contrast to the CO2 measurements from the CRDS, which have been studied thoroughly and have shown good performances, the quantitative capabilities of the CRDS water vapor measurements were never evaluated and reviewed in detail before. Precision in the laboratory is known from previous studies to be around 6 ppm for a 2.3 second integration time, but is related to white noise (Filges et al., 2015). Thus, sample averaging over 30 minutes can result in a precision of down to 0.3 ppm, which in principle can result in numerous scientific applications of the data. Each IAGOS aircraft is also equipped with the MOZAIC humidity device (Helten et al., 1998, Smit et al., 2008, Smit et al., 2013), which provides ..."

3) The frequent interchange between using ppm and % as units of mixing ratios (mole fractions) can lead to confusion, especially when it is desired to report relative differences or uncertainties (in percent). An example is Lines 5-6 on page 10 where "relative" is required to differentiate relative differences (in %) from mole fractions (in %). Mentally converting % to ppm is not terribly difficult, but by changing all the text and graph axes from % to ppm, you would completely avoid any potential confusion. Some of your 2-panel figures show mixing ratios in % (top panel) while differences or residuals are presented in ppm (bottom panel).

Mole fractions in unit "%" were changed to "ppm" in the whole manuscript (text, tables,

and figures). Ratios in "% (relative)" were changed to "%".

4) In many places uncertainties and error bars are presented without any indication if they are based on 1 or 2 standard deviations of the mean. It is helpful to report uncertainties, but only if the reader knows on which statistics they are based.

Added in section 3.1.2, page 7 line 30: "... dew point (2-sigma)) ..."

Added in section 3.1.3, page 8 line 9: "... 4 % (1-sigma) by ..."

Added in section 3.2.1, page 9 line 30: "... dew point (2-sigma). ..."

Added in section 3.2.1, page 10 line 28: "... 1.3 %, 2-sigma)..."

Added in section 3.2.3, page 11 line 28: "... 1.3 %, 2-sigma)..."

Added in section 3.2.3, page 12 line 5: "... uncertainty (1-sigma) ..."

Added in section 3.2.3, page 13 line 5: "... uncertainty (1-sigma) ..."

Added in section 4, page 14 line6: "... accuracy of $\pm0.1^\circ$C (1-sigma) dew point ..."

Added in section 4, page 14 line9: "... accuracy of 6-8 % (1-sigma) in ..."

Added in section 4.1.1, page 14 line26: "... measurement repeatability (1-sigma) of ..."

Added in section 4.1.1, caption of table 2: "... repeatability estimates (1-sigma) ..."

Added in Conclusions, page 17 line 23: "... 1.3 %, 2-sigma)..."

Added in Conclusions, page 17 line 26: "... 4 % uncertainty range (1-sigma) of..."

Added in Conclusions, page 17 line 29: "...uncertainty (1-sigma) ..."

Added in Conclusions, page 18 line 3: "...precision (1-sigma) ..."

Added in Conclusions, page 18 line 13: "Accuracy (1-sigma) of ..."

5) Please refrain from including Figure caption information in the main body text of the paper. It makes the paper more tedious to read. This occurs in numerous places throughout the manuscript.

Changed in Section 3.2.1, page 10 line 19: "... of the 2014-comparison."

Changed in Section 3.2.2, page 11 line 5: "... the data shows ..."

Changed in Section 3.2.3, page 11 line 25: "... the data indicates ..."

Removed in Section 4.1.1, page 14 line 27: "..., indicated by the three black lines in Fig. 7."

Removed in Section 4.1.2, page 15 line 9-10: "..., shown here as black points (30 seconds mean as grey points) ... (dark blue squares) ... (light blue triangles), ..."

Removed in Section 4.1.2, page 15 line 12-14: "..., are shown as orange diamonds; ..., are shown as green triangles ..."

Changed in Section 4.1.2, page 15 line 16: "... water vapor is added ..."

Changed in Section 4.1.3, page 15 line 26-28: "... water vapor and ... CR-2 and FISH, as well as ... pressure levels , for all four flights.

Removed in Section 4.1.2, page 15 line 28: For better comparison also the 30 seconds mean of the CRDS data is shown (in grey)."

Removed in Section 4.1.3, page line 2-7: "... (CRDS - CR-2 as black points, CRDS - FISH as dark blue diamonds, CR-2 - FISH as light blue triangles). The water vapor measurements of CR-2 are chosen as x-axis, because they cover all flights in contrast to the FISH data. The differences are calculated from the 30 seconds mean data of all analyzers and are averaged over intervals of 1 ppm, 10 ppm, 100 ppm, 1000 ppm, and 10000 ppm water vapor in the corresponding water vapor ranges of 0-10 ppm, 10-100 ppm, 100-1000 ppm, 1000-10000 ppm, and >10000 ppm, respectively. The standard deviations of the averaged differences are shown as error bars. For plotting reasons

all differences <1 ppm are set to 1 ppm. ..."

Added in caption of Figure 11: "... differences. The water vapor measurements of CR-2 are chosen as x-axis, because they cover all flights in contrast to the FISH data. For plotting reasons all differences <1 ppm are set to 1 ppm."

Removed in Section 4.1.3, page 16 line 16-20: "... (black points, 30 seconds mean as grey points) are (dark blue squares) ... (light blue triangles) ... (orange diamonds) ... (green triangles). ... (violet points) ..."

Changed in Section 4.1.3, page 16 line 29: "... temperatures below ..."

6) During the comparison flights, did all instruments sample the airstream from a common Rosemount TAT inlet, or did each have its own inlet? I believe only the FISH inlet is described as being different. If there was a common inlet for the other instruments, how was the sample stream physically split between them? If each instrument had its own inlet, how much of a difference might the different types and/or locations of inlets play in the overall comparisons? The absence of information about the sampling inlet for each instrument makes it very difficult to understand the inlet-dependent discussion on Page 16. I therefore suggest that this discussion be omitted from the paper because (a) it requires adding much greater detail about inlets and (b) inlet influences aren't necessarily a part of the evaluation of the CRDS measurement capabilities for IAGOS.

CR-2, Wasul-Hygro, and SEALDH I were connected to a common backward-faced inlet, FISH used a foreward-faced inlet.

Added in Section 4, page 14 line 9: "... (Meyer et al., 2015). The CR-2 was connected to a backward-faced inlet to avoid sampling of cloud and ice particles. In contrast, FISH measured ..."

Added in Section 4.1.3, page 16 line 19: "... as the CR-2. This is in line with expectation, since all three shared the same backward-faced inlet, which prevented from

sampling cloud droplets. However, the CRDS shows . . ."

7) The clarity of the presentation within this manuscript would be significantly improved through editing by a native English speaker. Hopefully that can be done before revisions are submitted.

Some proof reading was carried out.

Specific Comments:

1) Page 1, Line 16: "North-Germany" is not a proper noun, so "North" should not be capitalized and the hyphen should be omitted, i.e., "northern Germany". This occurs several times within the manuscript, e.g., Pg 4 L19.

Changed in Abstract (page 1, line 16): ". . .in northern Germany. . .".

Removed in Introduction (page 4, line 19): ". . .Four flights with a Learjet 35A took place in an area between North-Germany and South-Norway, and North-Poland and the North Sea respectively, reaching altitudes up to 12.5 km, hence covering also the lower stratosphere.. . ." (See answer to Specific comment 8)

Changed in section 4 (page 14, line 4): ". . .from northern Germany . . . to southern Norway. . .".

2) Pg 1, L19: What (specifically) is meant by "assessment validation"? Are there targets for the measurement precision and/or accuracy of this new IAGOS instrument that are the basis for its evaluation as described in this paper?

Changed in Abstract, page 1 line 19: ". . . for an initial validation of the . . ." See also answer to General Comment 2).

3) Pg 3, L6: "data of sufficient quality in the UTLS" - sufficient for what? This comment ties in with my previous comment about targets for "sufficient" measurement precision and/or accuracy.

Changed in Introduction, page 3 line 5-8: "... Solomon et al., 2010). However, due to only few existing measurement data in the UTLS, and imitations in prognostic model simulations of this region (Solomon et al., 2010), uncertainties in chemistry, transport processes, and trace gas composition are relatively large. This influences . . ."

4) Pg 3, L12-17: why include this long list of satellite-based water vapor sensors when their data are not included in this paper?

Changed in Introduction, page 3 line 13-15: ". . . (Rind et al., 1993). Recent observations are made by e.g. the Michelson Interferometer . . ."

Removed in References, page 20 line 10-12: Harries, J. E., Russell, J. M., Tuck, A. F., Gordley, L. L., Purcell, P., Stone, K., Bevilacqua, R. M., Gunson, M., Nedoluha, G., and Traub, W. A.: Validation of measurements of water vapor from the Halogen Occultation Experiment (HALOE), J. Geophys. Res., 101(D6), 10,205-10,216, 1996.

Removed in References, page 20 line 29-34: Lambert, A., Read, W., Livesey, N., Santee, M., Manney, G., Froidevaux, L., Wu, D., Schwartz, M., Pumphrey, H., Jimenez, C., Nedoluha, G., Cofield, R., Cuddy, D., Daffer, W., Drouin, B., Fuller, R., Jarnot, R., Knosp, B., Pickett, H., Perun, V., Snyder, W., Stek, P., Thurstans, R., Wagner, P., Waters, J., Jucks, K., Toon, G., Stachnik, R., Bernath, P., Boone, C., Walker, K., Urban, J., Murtagh, D., Elkins, J., and Atlas, E.: Validation of the Aura Microwave Limb Sounder middle atmosphere water vapor and nitrous oxide measurements, J. Geophys. Res., 112, D24S36, doi:10.1029/2007JD008724, 2007.

Removed in References, page 21 line 21-27: Read, W. G., Lambert, A., Bacmeister, J., Cofield, R. E., Christensen, L. E., Cuddy, D. T., Daffer, W. H., Drouin, B. J., Fetzer, E., Froidevaux, L., Fuller, R., Herman, R., Jarnot, R. F., Jiang, J. H., Jiang, Y. B., Kelly, K., Knosp, B. W., Kovalenko, L. J., Livesey, N. J., Liu, H.-C., Manney, G. L., Pickett, H. M., Pumphrey, H. C., Rosenlof, K. H., Sabounchi, X., Santee, M. L., Schwartz, M. J., Snyder, W. V., Stek, P. C., Su, H., Takacs, L. L., Thurstans, R. P., Vomel, H., Wagner, P. A., Waters, J. W., Webster, C. R., Weinstock, E. M., and Wu, D. L.: Aura Microwave Limb

Sounder upper tropospheric and lower stratospheric H2O and relative humidity with respect to ice validation, J. Geophys. Res., 112, D24S35, doi:10.1029/2007JD008752, 2007.

5) Pg 3, L18-19: "insufficient spatial resolution" - what spatial resolution is sufficient? And don't you instead mean "vertical resolution" here when discussing satellite and remote-sensing observations?

Added in Introduction, page 3 line 19: "... by clouds. As shown by Hoareau et al. (2013), vertical resolutions <500 m are needed for the simulation of cirrus clouds. To represent the very sharp gradient of 40 to 6 ppm water vapor within 0-2 km at the tropopause (Zahn et al., 2014), resolutions of even 400 m and higher have to be achieved (Poshyvailo et al., 2018). On the other hand ..."

Added in References, page 20 line 15: "Hoareau, C., Keckhut, P., Noel, V., Chepfer, H., and Baray, J.-L.: A decadal cirrus clouds climatology from ground-based and space-borne lidars above the south of France (43.9° N–5.7° E), Atmos. Chem. Phys., 13, 6951-6963, doi:10.5194/acp-13-6951-2013, 2013."

Added in References, page 21 line 21: "Poshyvailo, L., Müller, R., Konopka, P., Günther, G., Riese, M., Podglajen, A., and Ploeger, F.: Sensitivities of modelled water vapour in the lower stratosphere: temperature uncertainty, effects of horizontal transport and small-scale mixing, Atmos. Chem. Phys., 18, 8505-8527, doi:10.5194/acp-18-8505-2018, 2018."

Added in References, page 23 line 32: "Zahn, A., Christner, E., van Velthoven, P. F. J., Rauthe‐Schöch, A., and Brenninkmeijer, C. A. M.: Processes controlling water vapor in the upper troposphere/lowermost stratosphere: An analysis of 8 years of monthly measurements by the IAGOS‐CARIBIC observatory, J. Geophys. Res. Atmos., 119, 11505–11525, doi:10.1002/2014JD021687, 2014."

6) Pg 4, L6: "The central problematic of all these different ... are the remaining, unexplained discrepancies ...". Are the discrepancies "remaining"? From what? Are they a problem? It depends on the science you are investigating. Please improve the clarity of this statement.

Changed in Introduction, page 4 line 6: "... are the unexplained discrepancies ..."
Added in Introduction, page 4 line 11: "Thus, possibilities e.g. to identify long-term trends in stratospheric water vapor or to study ice microphysical processes are limited (Rollins et al., 2014)."

7) Pg 4, L15-18: This is the type of general statement that belongs in the abstract instead of the very detailed descriptions.

Lines 23-28 in the Abstract ("The inlet system, …. within IAGOS") were deleted.

8) Pg 4: I find a lot of the information presented in the abstract is duplicated here in the introduction. The introduction should explain why your investigation is important and how you intend to perform it. Specific information about instruments, including the MOZAIC humidity device, belongs in a separate section describing the various instruments.

Removed in Introction, page 4 line 18-20: "... Four flights with a Learjet 35A took place in an area between North-Germany and South-Norway, and North-Poland and the North Sea respectively, reaching altitudes up to 12.5 km, hence covering also the lower stratosphere. As reference instruments served ..."

Added in Introduction page 4 line 20: "The former were ..."

Added in Section 1, page 4 lines 28-29: "Therefore, a first impression of the quantitative capabilities of the CRDS water vapor measurements was needed, since they were never evaluated and reviewed in detail in a publication before. …"

9) Pg 5: There is a lot of repeated information on this page. Please ensure that you don't write the same sentence more than once in the paper, with the exception of abstract and conclusions.

Changed in Introduction, page 5 line 2-4: ". . . long performance record to evaluate . . ."

Changed in Introduction, page 4 line 21: ". . . Hygrometer (FISH) (Meyer et al., 2015) and a CR-2 Cryogenic Aircraft hygrometer (Buck Research Instruments L.L.C., Boulder, US, www.hygrometers.com), both . . ."

Changed in Introduction, page 5 line 9: ". . .flight tests are . . ."

10) Pg 5, L24: What "impact on gas density and spectroscopy" is being minimized? This statement is vague and needs clarification.

Changed in Section 2, page 5 line 24: "To minimize the impact of pressure and temperature on gas density and spectroscopy, both are kept constant in the sample cell."

11) Pg 6, L4-6: This description was already given in great detail, in the abstract (where it likely does not belong).

Lines 23-28 in the Abstract ("The inlet system, . . .. within IAGOS") were deleted. 12) Pg 6, L13: How are there only small humidity differences between the cabin and the outside air?

The cabin air is air conditioned. Therefore, differences in humidity between cabin and ambient air are small.

Added in Section 2, page 6 line 13: " . . . differences in humidity between the air conditioned cabin and the ambient air, . . ."

13) P6 6, L27-30: This would be the best place to explain that the instrument zero ("offset") is not stable (over what time scale?) and requires frequent adjustment, while the instrument gain is very stable and needs calibration only infrequently.

Added in Section 3.1.1, page 6 line 27: ". . . an offset correction is required once to improve .."

Added in Section 3.1.1, page 6 line 30: ". . . water vapor. The offset stability of different

CRDS instruments was checked regularly over a time period of up to 10 years and no significant drift was observed."

14) Pg 9, L30-32 and Figure 3: Are the reported uncertainties 1-sigma? 2-sigma? This comment applies to the entire paper, wherever uncertainties are given (text and Figures).

More details about the given uncertainties were added in the entire manuscript. See also answer to General Comment 3).

15) Pg 10, L1: "0.0024$\pm$0.0021" is "not significant"? What does "significant" mean here? It can't possibly mean "not statistically different from zero" as the reader might expect.

Changed in section 3.2.1, page 10 line 1: "The impact of the quadratic term determined as $0.0024 \pm 0.0021$, on the result is small compared to the overall uncertainty range. Thus, . . ."

16) Pg 10, L12-14, L20: The differences between calibration curves presented in ppm in Figures 3b and 4b and expressed in ppm and % (rel) in the text may cause confusion here. If you express all mixing ratios, differences and residuals in ppm it avoids any confusion. For the Figures 3b and 4b it would add value to include a right axis of relative differences/residuals. Given the near-linearity of the differences/residuals against mixing ratios, I think the relative data (different symbol/color) would be quite constant over the entire range of mixing ratios.

Mole fractions in unit "%" were changed to "ppm" in the whole manuscript (text, tables, and figures). Ratios in "% (relative)" were changed to "%".

A right axis of relative differences/residuals was added in Figures 3b and 4b.

Added in Section 3.2.1, page 10 line 5: "The relative difference of the two calibrations, shown on the right axis, increases . . ."

Added in Caption of Figure 3: "... calibration. The relative differences are plotted on the right axis as light blue triangles."

Added in Section 3.2.1, page 10 line 20: " The relative difference (right axis in plot (b)) between the two experiments is <0.3 % for water vapors <8000 ppm. Since ..."

Added in Caption of Figure 4: "...in plot (b). The relative residuals are plotted on the right axis as light blue triangles. Since the analyzers were not offset corrected before the experiment, the relative difference of 800 % of the data point at 4.5 ppm is less meaningful and is therefore not show in order to improve the clarity of the plot."

17) Pg 10, L26-28: Do you have enough evidence to make this statement based on a conjecture that the instrument "was not calibrated well enough" in 2009? This seems like a hand-waving statement with little factual backing. Is there further evidence that allows you to select the 2013 calibration as being correct?

Added and changed in Section 3.2.1, page 10 line 26: "... The dew point mirror has not been calibrated for nearly nine years when it was used for the 2009 experiment, but was not calibrated well during the 2013 experiment. Thus, ..."

18) Pg 10, L33: I have to assume that the mixing ratio ranges compared with each other instrument were chosen based on the best measurement ranges of the other instruments. Is this the case? If so, please inform the reader of this earlier in the paper. I see the range information in Table 1, but the table is not mentioned until page 13. An earlier statement about the choices of comparison ranges for different instruments and a mention of Table 1 are needed, otherwise it appears the comparison ranges were chosen randomly.

Added in Section 3.1.2, page 7 line 24: "... dew point. Higher and lower water vapor levels could not be reached due to the environmental conditions in the laboratory."

Added in Section 3.1.3, page 8 line 9: "... were measured. This corresponds to the standard calibration range of the FISH calibration bench and is a good addition to the

dew point mirror calibration range. Maximum ..."

Added in Section 3.1.4, page 9 line 12: "... constant. The adjustable water vapor levels were limited by the remaining humidity in the pressurized air on the one hand and the environmental conditions in the laboratory on the other. The ..."

Added in Section 3.2.4, page 13 line 27: "... experiments. The water vapor ranges used in the comparison were determined by the experimental setups of the experiments and the standard calibration ranges of the instruments. The ..."

19) Pg 11, L5-13 and Figure 5: The residuals in Figure 5b (presented in ppm) are discussed in the text in relative (%) terms. Adding a right axis for relative residuals to Figure 5b (using different symbol/color) would strengthen the discussion of relative residuals in the text.

A right axis of relative differences/residuals was added in Figures 5b.

Added in Section 3.2.2, page 11 line 8-10: "... residuals ... for water vapors > 100 ppm, which can be seen in plot (b), are small compared to the uncertainty range of the FISH calibration bench of 4 % indicated by the error bars. For the measurement point at 2 ppm water vapor the relative residuals are larger (6.2 %), due to the influence of the 12.2 ppm offset.

Added in Caption of Figure 5: "...of 4 %. The relative residuals are plotted on the right axis as light blue triangles."

20) Pg 11, L30-31: It is odd that the residuals at 250 ppm are negative (Figure 6b) while at the lowest mixing ratio they are positive. With an offset of 16 ppm one would expect the residuals at both of the lowest mixing ratios to be of the same sign. Any idea why they are negative at 250 ppm and positive at the lowest mixing ratio?

The residuals are negative at 2500 ppm, not 250 ppm, and positive at 300 ppm. Unfortunately there is only one data point at 300 ppm. With more measurements at this water vapor level we would expect the scatter/pattern of the residuals to be similar to

the ones at other water vapor levels. Therefore, as described in Section 3.2.3, the uncertainty range of 1.6 ppm of the offset is likely unrealistic and follow on experiments should contain more measurements at low water vapor levels (<300 ppm).

21) Pg 12, L1: If "the lowest measurement was made at around 300 ppm", why are there data points below 300 ppm in Figure 6?

There are no data points below 300 ppm. 1 % water vapor mole fraction corresponds to 10000 ppm, not 1000 ppm. For clarification all water vapor values were changed to ppm in the text and figures.

22) Pg 12, L6-7: "which was tried to exclude" is an example of where a native English speaker could help clean up the grammar in this manuscript.

Changed in Section 3.2.3, page 12 line 6-7: "..., which we tried to avoid by ..."

23) Pg 12, L10: It's not "zero air" if it contains 2 ppm CH4. You might want to use a different term.

Changed in Section 3.2.3, page 12, line 10: "...the pressurized air with 400 ppm CO2 ..."

24) Pg 12, L20: Of course, "the pressure sensor is sensitive to water vapor", as it is to any gas-phase constituent. I think you are trying to convey that the pressure sensor responds in a non-linear way to increasing amounts of water vapor, creating a water vapor-dependent bias in the pressure readings. In L25, what does "assumed that the pressure changes linearly" mean here? If the pressure changes linearly with changes in water vapor you don't have a problem with water vapor-dependent biases. I don't understand this assumption.

Added and changed in Section 3.2.3, page 12 line 20-22: "... the pressure sensor has a non-linear dependence on water vapor and thus, ... (Reum et al., 2017). A possible reason could be adsorption of water molecules on the sensor. To assess ... an incorrect pressure adjustment ..."

Added and changed in Section 3.2.3, page 12 line 25: "Experiments with an additional independent pressure measurement presented by Reum et al. (2017), as well as analysis of the behavior of the proportional valve, which controls the pressure in the sample cell, show that $\Delta p$ changes linearly with water vapor for mole fractions >2500 ppm (see Figure 2 in Reum et al., 2017). Therefore, the peak areas ... "

Removed in Section3.2.3, page 13 line 3: "Experiments with an additional independent pressure measurement (Reum et al., (2017), as well as analysis of the behavior of the proportional valve, which controls the pressure in the sample cell, show that" Changed in Section3.2.3, page 13 line 3: "Reum et al. (2017) determine $\Delta p$ as about 0.7 mbar ... Note however, that the change in cell pressure with humidity is not linear for water vapor mole fractions <2500 ppm, which could be the reason for the slightly systematic shape in the residuals at this low water vapor levels (Fig. 6 b).

25) Pg 12, L12-25: Don't the water vapor-dependent bias in the pressure readings also affect the in-flight data of the CRDS? Here you dwell on how this affects the CO2 dilution calibration method, but don't discuss the effects on the actual in-flight measurements. If this also affects the flight data, why only discuss it here in the "Calibration by CO2 Dilution" section?

For the in-flight measurements the error is removed, or rather corrected, by the calibration of the CRDS instrument with the dew point mirror. Moreover, the effect of the nonlinear part of the water dependent bias is small compared to the uncertainty range of the dew point mirror. The error due to the pressure deviation plays its role only during the dilution calibration, since here, the CO2 measurements of the same instrument, which are affected by this error as well, are used for the calibration.

Added in Section 3.2.3, page 12 line 21: "...While this error in the CRDS water vapor measurements is corrected by the calibration of the instrument with another hygrometer, it has to be considered for the dilution calibration, since the used CO2 measurements are affected as well. To ..."

26) Pg 13, L26: It is strange to have a section entitled "Summary" in the middle of this paper. How about "Calibration Summary" instead?

Changed in Section 3.2.4, page 13 line 26: "3.2.4 Calibration summary"

27) Pg 13, L30: "Note that both offsets, or rather their uncertainties, are likely not reliable." So, are the "offsets" (y-intercepts) likely not reliable, or their uncertainties, or both?

Changed in Section 3.2.4, page 13 line 30: "... both offset uncertainties are ..."

28) Pg 14, L8-9: What is "a total accuracy" and what does "and down to 0.3 ppm for the lower mixing ratios" mean? How low is "lower"?

Changed in Section 4, page 14 line 8-9: "... has an accuracy of 6-8 % (1-sigma) in the range from 4 to 1000 ppm and to 0.3 ppm for lower mixing ratios down to 1 ppm (Meyer et al., 2015)."

29) Pg 14, L14-15: What are the "stable atmospheric conditions" mentioned here? This sentence needs a comma after "analysers".

Added in Section 4.1.1, page 14 line 14-15: "analyzers, water ... atmospheric conditions, as pressure and temperature, were ..."

30) Pg 14, L13: Why use the term "repeatability" instead of precision? From the subsequent description it appears that you are looking at variations in the 0.4 Hz measurements from the 60-second averages. I would call this "precision" and not repeatability, since repeatability can imply stability in results obtained at different times, such as re-sampling the same air mass five minutes later and seeing how "repeatable" the measurements are.

Changed in Section 4.1.1, page 14 line 13: "Measurement precision" Changed in Section 4.1.1, page 14 line 14: "... measurement precision ..." Changed in Section 4.1.1, page 14 line 16: "... the precision can be..." Changed in Section 4.1.1, page 14 line

26: "... the measurement precision ..." Changed in Section 4.1.1, page 14 line 28: "For comparisonprecision estimates ..." Changed in Section 4.1.1, page 15 line 1: "... a precision of ..." Changed in Section 4.1.1, page 15 line 3: "... the precision of the ..." Changed in Table 2, caption: "Precision estimates ..." Changed in Table 2: "precision at ..." Changed in Section 3.2.1, page 10 line 10: "Precision of the CRDS analyzer ..."

31) Pg 14, L18-19: Isn't the 60-second average calculated from the 2.5-second measurements? Then why do you calculate the standard deviation of their differences when the standard deviation of the 60-second average directly provides a direct statistical measure of variability in the shorter-term measurements?

The 60-second moving average is a measure for the long-periodic variability. Subtracting this long-periodic part of the signal from the measurements leaves the short-periodic variability, which is our measure for the short-term precision.

32) Pg 14, L24: "results based on a larger number of data are highlighted". Aren't most 60-second averages based on 24 measurements made at 0.4 Hz?

Only the flight measurement data from the four flights (total flight time: 14 h) are used for the analysis. Since not all water vapor ranges were represented equally during the flight measurements and, moreover, data periods with unstable cell pressure had to be neglected, the data availability in the different water vapor intervals varied significantly.

33) Pg 14, L26-27: "upper limit" ... "4 ppm or 5%, whichever is greater for WV < 100 ppm". In Figure 7, why are there so many data points for WV < 80 ppm that have standard deviations > 4 ppm (5%)? The upper limit estimate for WV > 100 ppm is much more in line with the Figure 7 data: most of the data points for WV > 100 ppm lie below the 30 ppm standard deviation line and to the right of the 5% diagonal. This cannot be said about the WV < 100 ppm data, many of which lie above the 4 ppm standard deviation line and to the left of the 5% diagonal. Please explain the connection between this Figure and the numbers you have extracted from it that appear to underestimate the precision values for measurements at WV < 100 ppm.

See answer to General Comment 1).

34) Pg 15, L9-16: No need to describe the different symbols since that is done in both the Figure 8 caption and legend. What is more important here is a description of what the Figure shows about response time of the CRDS. I think that showing only a short period of Figure 9 would better support this discussion of response times <10 seconds. Specifically, the flight segment from 12.35 to 12.55 UTC on 1 June 2011 when water vapor increases and falls an order of magnitude in about 3 minutes.

See also answer to General Comment 5).

Enlarged sections for selected shorter time periods are added in Figure 8 and 9.

Added in Caption of Figure 8: "... flight on 31 May 2011. An enlarged section for the time period from 11:15 to 11:24 is shown in the lower left part of the plot."

Added in Caption of Figure 9: "...flight on 1 June 2011. An enlarged section for the time period from 12:24:36 to 12:25:48 is shown in the upper left part of the plot. Here the water vapor increases in one minute about an order of magnitude from 200 to 1200 ppm."

Added in Section 4.1.2, page 15 line 19: "... to wet. For water vapors <100 ppm the response time is comparable to the FISH instrument, as shown in the enlarged section of Fig. 8. During the increase of water vapor from 200 to 1200 ppm in about one minute, shown in the enlarged section in Figure 9, no significant delay can be detected. Thus, ..."

35) Pg 15, L22-25: This discussion of laboratory tests of response time is not very informative or conclusive because the reader has no idea how much water vapor is in the "wet and dry air". Also, "recovery time to 99% of a challenge" is not understandable.

Added in Section 4.1.2, page 15 line 22: "... between wet (around 23000 ppm) and dry (around 10 ppm) air, ..."

Changed in Section 4.1.2, page 15 line 23: "... to 99 % of the final water vapor level as ..."

Changed in Section 4.1.2, page 15 line 24: "... from 23000 ppm to ..."

36) Pg 15, L27-31 and Pg 16 L1-6: Please remove Figure 11 symbol descriptions and the explanation of data binning since all this information appears in the Figure caption.

Descriptions and explanations were removed. See answer to General Comment 5.

37) Pg 16, L1-2 and Figure 11 caption: "Absolute" differences steers my thinking towards absolute values, which are definitely not what Figure 10 shows. "Absolute" is unnecessary in both instances because the differences are clearly in units of ppm, i.e., not relative differences.

Changed in Section 4.1.3, page 16 line 1: "The water vapor differences ..."

Changed in Figure 11, caption: "Differences ..."

38) Pg 16, L4-6 and Figure 11: If the differences are binned by CR-2 water vapor in 1 ppm bins for CR-2 < 10 ppm, why are there multiple blue diamonds and multiple blue triangles in the lowest CR-2 mixing ratio bin (and some other bins)? Aren't the averages computed from all segments of all flights? If not, why are the average differences in each bin determined for subsets of the 4 flights?

Figure 11 was corrected. Now there is only one data point for each bin.

39) Figure 11: The log-scale Y-axis is problematic for average differences < 1 ppm and for the error bars that appear asymmetric around the mean. Why don't you instead plot average relative (%) differences using a linear-scale Y axis? Average differences < 1 ppm will then be displayed correctly and the error bars will be symmetric around the mean values. I think the relative values will be much more like flat lines with CRDS-CR2 values above zero and the other instrument pairs below zero.

We plotted the relative differences using a linear axis, too. The problem with this display

is that for water vapors <100 ppm, and especially <10 ppm, the relative differences are very large (up to 50-100 %). For the interpretation of the results at <100 ppm a plot with differences in ppm is much clearer and easier to understand. So, both representations have their pros and cons. In the end we decided that the plot with the log-scales is a bit better suited for our purposes.

40) Pg 16, L11-33: This section is difficult to understand because there are no descriptions of the various inlets for the different instruments. All that is known at this point is that the FISH inlet is different from the CRDS inlet, although there is now a hint that the other instruments do not share the CRDS inlet. Is it really important or necessary to include this section of the paper when it requires adding more details of the various instrument inlets? This section seems to cover a different topic from the main thrust of this paper and may even be worthy of a separate paper in the future.

See answer to General Comment 6).

Please also note the supplement to this comment:
https://www.atmos-meas-tech-discuss.net/amt-2018-36/amt-2018-36-AC1-supplement.pdf
* * *
[Figure]

**Fig. 1.** fig03_dew point mirror calibration

[Figure]

**Fig. 2.** fig04_waterscale adaption of two Picarros

[Figure]

**Fig. 3.** fig05_FishCalBench calibration

[Figure]

**Fig. 4.** fig06_dilution calibration

[Figure]

**Fig. 5.** fig07_measurement precision

[Figure]

**Fig. 6.** fig08_response time wet-dry

[Figure]

**Fig. 7.** fig09_response time dry - wet

[Figure]

**Fig. 8.** fig11_absolute differences_all flights

**Supplement:**

[revised manuscript text omitted]

---

## Author Comment (AC2) · 21 Jul 2018

Author's response to Interactive comment by Anonymous Referee #2 on "Evaluation of the IAGOS-Core GHG Package H2O measurements during the DENCHAR airborne inter-comparison campaign in 2011" by Annette Filges et al.

A version of the manuscript with all changes is added as supplement to the author's response to referee #1.

All changed Figures are added in the author's response to referee #1.

General Comments

1) The manuscript would benefit substantially from copyediting to correct usage and punctuation. Examples include: the use of commas after introductory prepositional phrases,

Some proof reading was carried out.

2) The inclusion of water vapor mixing ratios in both ppm (ppmv) and % units is somewhat awkward, although I understand that those units are regularly used for water in different environments. Suggest consistently using ppmv for the manuscript with an initial parenthetical equivalence for the reader, e.g. on line 30 "7000 ppmv (0.7%) to 25,000 ppmv (2.5%)". This would have the additional benefit of simplifying the discussion of the uncertainty in % of measurement without needing to specify "(rel.)". The DENCHAR website uses ppmv even for high values mentioned.

Mole fractions in unit "%" were changed to "ppm" in the whole manuscript (text, tables, and figures). Ratios in "% (relative)" were changed to "%".

3) The details of representations (e.g. blue line, dark blue triangle) of data in the plots should not be included in the text, only in the figure captions (where they should be fully described for clarity). The text should discuss the interpretation and implications of the data presented in the figures. The figure captions could be more descriptive.

Text and figure captions were changed accordingly. See also answer to General Comment 5) of referee #1.

4) The term "repeatability" is used in a number of places in the manuscript (e.g. section 4.1.1) that would appear to be better described as measurement precision. Repeatability is typically the measure of how consistently an instrument will produce the same response when challenged at different times with the same inputâĚŸAĚĞTrelated to drift. Precision is more typically used for the short time variation of measurement (as is the case here for the variation about the 60 s or 30 s mean value) or repeated consecutive measurements of a constant input.

"Repeatability" was replaced by "precision". See answer to Specific comment 30) of referee #1.

5) Some pieces of information are repeated numerous times within the manuscript out of a desire to be clear, but it seems to go too far. For example, it should be stated at the first reasonable opportunity in the manuscript that 0.802 calibration factor from 2013 will be used going forward and then it need not be repeated every time.

Changed in Section 3.2.2, page 11 line 1-2: " . . . 2013 dew point mirror comparison, . . ."

Changed in Section 3.2.3, page 11 line 25: " . . . 2013 dew point mirror comparison, . . ."

Changed in Section 4.1.1, page 14 line 16-17: " . . . (according to the 2013 dew point mirror comparison; in the following simply referred to as CRDS measured water vapor) . . ."

Removed in Section 4.1.2, page 15 line 7-8: "In addition to the offset corrected and calibrated (according to the 2013 dew point mirror comparison, calibration factor = 0.802 ± 0.010) water vapor measurements of the CRDS analyzer, in the following simply referred to as . . ."

Specific Comments:

1) P1, L16: "North-Germany" should be "northern Germany"; similarly elsewhere, "South- Norway" and "North-Poland" should be "southern Norway" and "northern Poland"

Changed in the whole manuscript. See answer to Specific Comment 1) of referee #1.

2) P1, L23: This paragraph seems inappropriate for the abstract–suggest removing and including the details in the instrument description section.

Lines 23-28 in the Abstract ("The inlet system, . . . . within IAGOS") were deleted.

[Figure]

Added in Section 2, page 6 line 11: "operation of the instrument with a controlled pressure in the sample cell of 186.65 hPa (140 Torr) throughout the aircraft altitude operating range..."

3) P1, L30: "instruments" should be capitalized.

Changed in Abstract, page 1 line 30: "...Michell Instruments Ltd. ..."

4) P1, L31+: "on the ground over the range from 2 to 600 ppmv against the dew point hygrometer used for calibration of the FISH instrument. A new, independent calibration method based on the dilution effect of water vapor on CO2 was evaluated."

Changed in Abstract, page 1 line 31-33: "...compared on the ground over the range from 2 to 600 ppm against the dew point hygrometer used for calibration of the reference instrument FISH. A new, independent calibration method based on the dilution effect of water vapor on CO2 was evaluated."

5) P2, L3: "than" should be "then", but even better could just be deleted–"1% for the water vapor range from 25,000 ppmv down to 7000 ppmv, increasing to 5% at 50 ppm"

Changed in Abstract, page 2 line 3: "... down to 7000 ppm, increasing to ..."

6) P3, L6: sufficient quality for what?

Changed in Introduction, page 3 line 5-8: "... Solomon et al., 2010). However, due to only few existing measurement data in the UTLS, and ..."

7) P3, L7: "difficulty representing"; overall this sentence is awkward and difficult to parse.

Changed in Introduction, page 3 line 5-8: "However, due to only few existing measurement data in the UTLS, andlimitations in prognostic model simulations of this region (Solomon et al., 2010), uncertainties in chemistry, transport processes, and trace gas composition are relatively large. This influences ..."

8) P3, L18: for what purpose do the satellite measurements have insufficient spatial resolution? Horizontally? Vertically?

See answer to Specific Comment 5) of referee #1.

9) P3, L32: What is meant by "beneath"?

Changed in Introduction, page 3 line 32: "Besides frost point hygrometers . . ."

10) P5, L28: "independent on ambient, respectively cabin pressure" is confusing. "independent of both ambient and cabin pressure"?

Changed in Section 2, page 5 line 28: ". . . ambient and cabin pressure . . ."

11) P5, L29: the filters in the sample line are likely contributing significantly to the long tail in the time constant.

The physical exchange time of the sample cell is only 3.6 s (volume = 35 ml, sample flow = 100 ml/min, pressure = 186.65 hPa, sample temperature = 45°C). Surface effects on the walls of the inlet line and sample cell, as well as the filters can contribute to a longer time constant, especially when the change in water vapor is large. However, comparison of all instruments in Section 4.1.2 shows that the response time of the CRDS is similar to that of the other instruments.

12) P7, L19: the use of _125 ppm as a metric for stability over a range that includes 3 ppm does not induce confidence–would be better to use a relative measure.

Changed in Section 3.1.2, page 7 line 19: ". . . ranging from around 5000 to 30000 ppm." After correcting the offsets of all analyzers we expect the relative differences for water vapor <5000 ppm to also be around some percent.

13) P8, L15: "the permeation is negligible" for 3500 sccm is technically incorrect since the permeation rate through the tubing is largely flow independent. The contribution of the permeation to the water vapor mixing ratio in the flow might be argued to be negligible.

Added in section 3.1.3, page 8 line 15: "... the contribution of the permeation to the water vapor mole fraction in the flow is negligible ..."

14) P9, L5: "gets only changed by" → "is only affected by" Changed in Section 3.1.4, page 9 line 5: "... the peak area gets only changes due to the dilution effect. ..."

15) P9, L13: "whereby it required about" → "requiring"

Changed in Section 3.1.4, page 9 line 13: "...line, requiring about 1.3 s ..."

16) P9, L29: it would be helpful to the reader to convert the °C uncertainty of the Dewmet to mixing ratio (using the relevant operating pressure) for comparison with other uncertainties.

Added in Section 3.2.1, page 9 line 30: "... dew point (2-sigma), which corresponds to a relative uncertainty of 1.3 %."

17) P11, L11: the CRDS and FISH cal system can't agree within 3% at mixing ratios below 400 ppm without first correcting for the 12.2 ppm offset. It seems reasonable that there was either an outgassing or small leak (it doesn't take much) that led to the offset, but that should be argued and removed before stating the agreement in the response (slope).

Added in Section 3.2.2, page 11 line 11: "... up to 600 ppm after correcting for an offset of 12 ppm. ..."

Added in Section 3.2.2, page 11 line 21: "...than 1 ppm. Another possibility would be that the offset was caused by either an outgassing or a very small leak."

18) P12, L3: "likely unrealistic"–should be more descriptive; "the calculated error of 1.6 ppmv for the offset is likely a significant underestimate."

Changed in Section 3.2.3, page 12 line 3: "...is likely a significant underestimate."

19) P12, L20: The section is unclear–are you saying that the pressure sensor response

to H2O is different from that to other molecules, leading to an incorrect measurement of the wet flow cell pressure that depends on the H2O content?

See answer to Specific Comment 24) of referee #1.

20) P13, L4: It would be better to state a best estimate of the CO2 dilution method uncertainty rather than saying that a value "is achievable". What do you estimate that you actually achieved, and is that consistent with the comparisons?

The conservative estimate for the presented experiment is 1 % (see page 13, line 5). For future experiments uncertainties at sub-percent level are achievable. Added in Section 3.2.3, page 13 line 4: "... is achievable for the dilution method in future experiments."

21) P14, L2: "Hohn, Germany, the flights"

Changed in Section 4, page 14 line 2: "... Hohn, Germany, the flights ..."

22) P14, L5: "Hence, also the lower stratosphere was reached." is an awkward construction, this could be better included within the previous sentence.

Changed in Section 4, page 14 line 5: "... up to 12.5 km, so that also the lower ..."

23) P14, L10: "total water instead of only water vapor since the forward-oriented inlet resulted in sampling of cloud droplets and ice crystals when present." The sentence "Both instruments were..." seems unnecessary.

Changed in Section 4, page 14 line 10: "...instead of only water vapor ..., since it's forward-oriented inlet resulted in sampling of cloud droplets and ice crystals when present.

Removed in Section 4, page 14 line 10: "Both instruments were operated ba the research centre Jülich."

24) P14, L14: "CRDS analyzer, water vapor measurements during periods with stable"

Changed in Section 4.1.1, page 14 line 14: ". . . CRDS analyzer, water vapor . . ."

25) P14, L26: the value of 4 ppm/5% stated for the <100 ppmv precision seems to be at odds with the plotted standard deviations in the 10-100 ppmv range, which are typically higher than both.

See answer to General Comment 1) of referee #1.

26) P15, L9: "and the flight data. . ." → "along with the flight data from the reference instruments CR-2 and FISH. Water vapor measurements from two additional analyzers that participated in the intercomparison campaign are also shown."

Changed in Section 4.1.2, page 15 line 8-11: "CRDS measured water vapor is shown along with the flight data from the reference instruments CR-2 and FISH. Water vapor measurements from two additional analyzers that participated in the inter-comparison campaign are also presented: . . ."

27) P15, L17: "not slower than any other instrument." → "is similar to that of the other instruments."

Changed in Section 4.1.2, page 15 line 17: ". . .is similar to that of the other instruments."

28) P15, L30: "CRDS data from the first flight" and "influenced" → "compromised"

Changed in Section 4.1.3, page 15 line 30: ". . . CRDS data from the first flight . . . were compromised by . . ."

29) P16, L1: "can be seen" → "are shown" or "are presented" or "are plotted"

Changed in Section 4.1.3, page 16 line 1: ". . . water vapor intervals are plotted in . . ."

30) P16, L7: "are neglected." → "are omitted."

Changed in Section 4.1.3, page 16 line 7: ". . .are omitted. . . ."

31) P16, L11: "sampled data only for two flights" → "operated successfully for only two

of the four flights"

Changed in Section 4.1.3, page 16 line 11: "... FISH operated successfully for only two of the four flights, ..."

32) P16, L20: "This might indicate that also the CRDS samples cloud" → "This indicates that the CRDS sampling was likely also affected by cloud particles" Looking at the Rosemount inlet, there is a good likelihood that impaction of droplets or ice crystals (shattering) will produce particles small enough to be entrained in the sample flow and ingested into the instrument. The ice crystal fragments would then be small enough to sublimate in the inlet and affect the measurement.

Changed in Section 4.1.3, page 16 line 20-21: "This indicates that the CRDS sampling was likely also affected by cloud particles, ..."

Added in Section 4.1.3, page 16 line 26: "Such small enough particles could be produced e.g. by the shattering of water droplets or ice crystals in the Rosemount housing."

33) P18, L1: It is unclear exactly what is meant by the statement "The dilution method can be used for other species...". The dilution method could be used to determined $H_2O$ using methane instead of $CO_2$, or are you trying to say that methane could be measured by the method by comparing to $CO_2$?

Both options are meant. $H_2O$ can be calibrated using another species instead of $CO_2$, and other species instead of $H_2O$ can be also calibrated by $CO_2$ or any another species, provided both used species are measureable by a CRDS analyzer and the dilution effect is large enough.

Changed in Conclusions, page 18 line 1: "... used for the calibration of other species, too, provided they and the corresponding diluted species are measurable by a CRDS analyzer and the dilution effect ..."